# Thermosensitive alternative splicing senses and mediates temperature adaptation in *Drosophila*

Ane Martin Anduaga[1†], Naveh Evantal[2†], Ines Lucia Patop[1], Osnat Bartok[2], Ron Weiss[2], Sebastian Kadener[1,2]*

[1]Biology Department, Brandeis University, Waltham, United States; [2]Silberman Institute of Life Sciences, The Hebrew University of Jerusalem, Jerusalem, Israel

**Abstract** Circadian rhythms are generated by the cyclic transcription, translation, and degradation of clock gene products, including *timeless* (*tim*), but how the circadian clock senses and adapts to temperature changes is not completely understood. Here, we show that temperature dramatically changes the splicing pattern of *tim* in *Drosophila*. We found that at 18°C, TIM levels are low because of the induction of two cold-specific isoforms: *tim-cold* and *tim-short and cold*. At 29°C, another isoform, *tim-medium*, is upregulated. Isoform switching regulates the levels and activity of TIM as each isoform has a specific function. We found that *tim-short and cold* encodes a protein that rescues the behavioral defects of *tim^01* mutants, and that flies in which *tim-short and cold* is abrogated have abnormal locomotor activity. In addition, miRNA-mediated control limits the expression of some of these isoforms. Finally, data that we obtained using minigenes suggest that *tim* alternative splicing might act as a thermometer for the circadian clock.

*For correspondence:
skadener@brandeis.edu

†These authors contributed equally to this work

Competing interests: The authors declare that no competing interests exist.

## Introduction

Circadian rhythms organize most physiological and behavioral processes into 24 hr cycles (*Allada and Chung, 2010*; *Pilorz et al., 2018*). The current model postulates that circadian clocks keep time through a complex transcriptional-translational negative feedback loop that takes place in the so-called 'clock cells' (*Shafer et al., 2006*; *Helfrich-Förster, 2003*). It has been proposed that each clock cell functions autonomously (*Dissel et al., 2014*; *Yoshii et al., 2012*). In *Drosophila*, the master regulators, CLOCK (CLK) and CYCLE (CYC), activate the circadian system by promoting the rhythmic transcription of several key clock genes. The products of three of these genes, PERIOD (PER), TIMELESS (TIM), and CLOCKWORK ORANGE (CWO), repress CLK-CYC-mediated transcription in an oscillatory manner (*Zhou et al., 2016*; *Hardin and Panda, 2013*). These cycles of transcriptional activation and repression lead to 24-hr molecular oscillations, which ultimately generate behavioral rhythms. In addition to transcriptional control, post-transcriptional and post-translational regulatory processes are essential for circadian timekeeping (*Lim and Allada, 2013*; *Harms et al., 2004*; *Hardin and Panda, 2013*; *Kojima et al., 2011*). For example, the transcriptional repressors PER and TIM are post-translationally modified; modification status and the rates with which these modifications take place have a significant influence on the degradation of these proteins (*Ozkaya and Rosato, 2012*; *Hardin and Panda, 2013*). Modification of PER may also influence its transcriptional repressor activity or the timing of its activity (*Ozkaya and Rosato, 2012*; *Hardin and Panda, 2013*).

Circadian clocks are extraordinarily robust systems; they are able to keep time accurately without timing cues. In addition, and despite their biochemical nature, they are resilient to large variations in environmental conditions. The robustness of the circadian system is probably the result of multiple layers of regulation that ensure accurate timekeeping and the buffering of stochastic changes in the

molecular clockwork. These levels of regulation are physically and/or functionally interconnected and, importantly, they extend beyond the single-cell level. Circadian neurons in the brain are organized in a network that is believed to synchronize the individual neuronal oscillators, thereby contributing to a coherent and robust behavioral output (*Mezan et al., 2016*; *Weiss et al., 2014*; *Stoleru et al., 2004*; *Peng et al., 2003*; *Schlichting et al., 2016*; *Allada and Chung, 2010*). The neuropeptide PDF, which is the main neuromodulator of the circadian neuronal network, is expressed in a small subset of these neurons. PDF is essential for normal activity patterns in light: dark (LD) cycles, as well as for persistent circadian rhythms in constant darkness (DD) (*Peng et al., 2003*; *Im et al., 2011*; *Taghert and Shafer, 2006*). Circadian clocks are also remarkably plastic systems. For example, organisms can quickly adjust to different light and temperature regimes (*Roessingh et al., 2015*; *Wolfgang et al., 2013*). The plasticity of the circadian clock results from the existence of very efficient input pathways that can convey external signals into the core oscillator machinery (*Bartok et al., 2013*; *Ogueta et al., 2018*).

TIM is at the crossroads of the robustness and plasticity of the circadian clock. TIM is a core circadian oscillator component, and mutations in *tim* cause flies to have short, long, or no circadian rhythms (*Rothenfluh et al., 2000*; *Sehgal et al., 1994*; *Myers et al., 1995*). TIM stabilizes PER, which is also essential for circadian rhythmicity, and this complex represses CLK-mediated transcription in an oscillatory manner (*Hall, 2003*). The stoichiometric relationship between PER, CLK, and TIM is tightly controlled and is probably the major regulator of circadian period (*Hardin, 2011*; *Yu et al., 2006*; *Kadener et al., 2008*; *Fathallah-Shaykh et al., 2009*). In addition, TIM is a key factor for communicating external information to the central core oscillator. For example, upon light stimulus, the protein encoded by *cryptochrome*, CRY, binds to TIM and promotes TIM degradation through the ubiquitin ligase Jetlag, which results in phase advances or delays of the circadian oscillator (*Lin et al., 2001*; *Koh et al., 2006*; *Emery et al., 1998*). PDF signaling has also been proposed to synchronize the circadian clock by regulating TIM degradation (*Seluzicki et al., 2014*; *Li et al., 2014*).

Temperature has diverse effects on the *Drosophila* circadian system (*Kidd et al., 2015*; *Afik et al., 2017*). Flies show temperature-dependent changes in the distribution of daily locomotor activity, known as seasonal adaptation (*Majercak et al., 1999*; *Low et al., 2008*). Flies can also be entrained by daily temperature cycles (*Roessingh et al., 2015*) and are phase-shifted by temperature pulses or steps (*Glaser and Stanewsky, 2005*). Flies keep 24 hr periods in a wide range of temperatures, a phenomenon known as temperature compensation (*Kidd et al., 2015*). The adaptation of *Drosophila* to different environmental conditions relies on the presence of at least two neuronal circuits: one that controls the morning component of locomotor activity (M) and another that controls the evening (E) burst of activity (*Stoleru et al., 2005*; *Stoleru et al., 2004*; *Grima et al., 2004*). The environment fine-tunes the activity pattern by altering the timing of these oscillators. In the laboratory, mimicking summer by subjecting the flies to hot temperatures and long photoperiods shifts their M activity to pre-dawn and their E activity into the early night (*Majercak et al., 1999*). By contrast, under conditions that mimic autumn (shorter day lengths and cooler temperatures), the M and E activity components fall closer together and occur around the middle of the day.

In 1999, work from the Edery lab first linked the splicing of the *per* 3′ untranslated region (UTR) to seasonal adaptation (*Majercak et al., 1999*). The evening advance in locomotor activity at lower temperatures correlates with an advance in the phase of oscillation of *tim* and *per* mRNAs. The authors postulated that this shift of the clock is driven by the alternative splicing of an intron located in the *per* 3′ UTR (dmpi8 intron). Further studies revealed that the efficiency of splicing of this intron is regulated not only by temperature but also by the photoperiod (*Collins et al., 2004*). The effect of light on *per* splicing does not require a functional clock, and phospholipase C plays a physiological, non-photic role in regulating the production of the alternatively spliced transcript (*Collins et al., 2004*; *Majercak et al., 2004*). In addition, the splicing factor B52 (also called SRp55) binds to *per* 3′ UTR and modulates the splicing of the dmpi8 intron (*Zhang et al., 2018*). A study published in 2008 examined the splicing pattern of *per* in the tropical species *Drosophila yakuba*, which faces no significant seasonal variation in day length or temperature (*Low et al., 2008*; *Ko et al., 2003*; *Russo et al., 1995*). The *per* gene in *D. yakuba* has a 3′-terminal intron. This intron is removed over a wide range of temperatures and photoperiods, consistent with the marginal effect of temperature on this tropical species. Despite these findings, it is still not clear how the regulation of the 3′ UTR splicing impacts PER levels, or whether other transcriptional and/or post-transcriptional events may

also influence the response to temperature changes. A recent report suggests that the effect of the *per* dmpi8 intron on midday siesta might be PER-independent, and that splicing of the *per* 3′ UTR acts in trans to regulate the expression of *daywake* (*dyw*), which encodes an anti-siesta factor (*Yang and Edery, 2019*).

Post-transcriptional control by miRNAs and RNA-binding proteins is important for circadian time-keeping in flies and other organisms (*Lerner et al., 2015*; *Kadener et al., 2009*; *Lim and Allada, 2013*; *Xue and Zhang, 2018*; *Chen and Rosbash, 2016*). In addition, several recent reports show a role for alternative splicing in circadian timekeeping (*Petrillo et al., 2011*; *Bartok et al., 2013*; *Sanchez et al., 2010*; *Wang et al., 2018*). This is not surprising, given the prevalence of alternative splicing in the brain and the importance of this process in regulating the amount and type of mRNAs that are generated from a given locus (*Baralle and Giudice, 2017*; *Sanchez et al., 2011*). Alternative mRNA processing is co-transcriptional and usually happens because of the presence of suboptimal processing signals (i.e., non-canonical splice sites or polyadenylation sites). Many factors can influence alternative splicing, including RNA structure, RNA-binding proteins, and the elongation rate of RNA polymerase II (*de la Mata et al., 2003*; *Cáceres and Kornblihtt, 2002*).

Here, we show that temperature dramatically and specifically changes the splicing pattern of *tim*. We found that the reduced levels of the canonical TIM protein at 18°C result from the induction of two cold-specific splicing isoforms: *tim-cold* and *tim-short and cold* (*tim-sc*). *tim-cold* encodes a canonical TIM protein, but expression from this mRNA is under strong miRNA-mediated control, as shown by ARGONAUTE 1 (AGO1) immunoprecipitation and in vivo luciferase reporter assays. *tim-sc* encodes a short TIM isoform that can advance the phase of the circadian clock when overexpressed and that is able to partially rescue *tim01* flies. In addition, we established that TIM-SC binds to but does not stabilize PER. We generated flies that do not produce *tim-sc* using CRISPR technology. These flies display altered patterns of locomotor activity at 18°C and at 25°C, as well as altered expression of the other *tim* isoforms. Interestingly, most of the changes in *tim* splicing patterns are conserved across *Drosophila* species and correlate with the capacity of the species to adapt their activity to temperature changes. Moreover, we showed that the temperature-dependent changes in *tim* alternative splicing are independent of the circadian clock. These findings were reproduced using minigenes that contain the temperature-sensitive introns and two flanking exons in *Drosophila* S2 cells. The latter results strongly suggest that the intronic sequences of *tim* themselves are the temperature sensors that lead to changes in splicing that result in adaptation to a wide range of temperatures.

## Results

### Temperature remodels the circadian transcriptome

To analyze the effect of temperature on general and circadian gene expression, we evaluated gene expression genome-wide in the heads of flies maintained at 18°C, 25°C, or 29°C. Briefly, we performed 3′ RNA-seq at six time points in flies entrained to 12:12 LD cycles at these three temperatures. We identified hundreds of genes that had oscillating expression over the day (*Figure 1A* and *Supplementary file 1*). Interestingly, most of the cycling RNAs were temperature specific: only a few RNAs had oscillating expression at all of the assayed temperatures (*Figure 1B* and *Supplementary file 1*). The sets of mRNAs that oscillated at each temperature were enriched for different Gene Ontology (GO) terms, demonstrating that the circadian program is temperature dependent (*Supplementary file 2*). Interestingly, we observed a phase advance in several core circadian components at 18°C relative to 25°C or 29°C (*Figure 1C*), suggesting that the circadian clock gene expression program is advanced at low temperature. The phase advance mirrors the advance of the evening activity component observed at this lower temperature. Despite the clear behavioral differences, we observed smaller phase changes in the expression of circadian clock genes in flies entrained at 29°C compared to those entrained at 25°C than the observed between 25°C and 18°C (*Figure 1C* and *Supplementary file 1*). As previously described (*Majercak et al., 1999*), *tim* and *per* mRNA profiles were advanced at 18°C compared to those at 25°C. This phase advance is transcriptional, as shown by analysis of chromatin-bound *tim* and *per* mRNAs (*Figure 1—figure supplement 1*). This phase advance in the mRNA levels provokes a phase advance in TIM and PER protein levels (*Majercak et al.,*

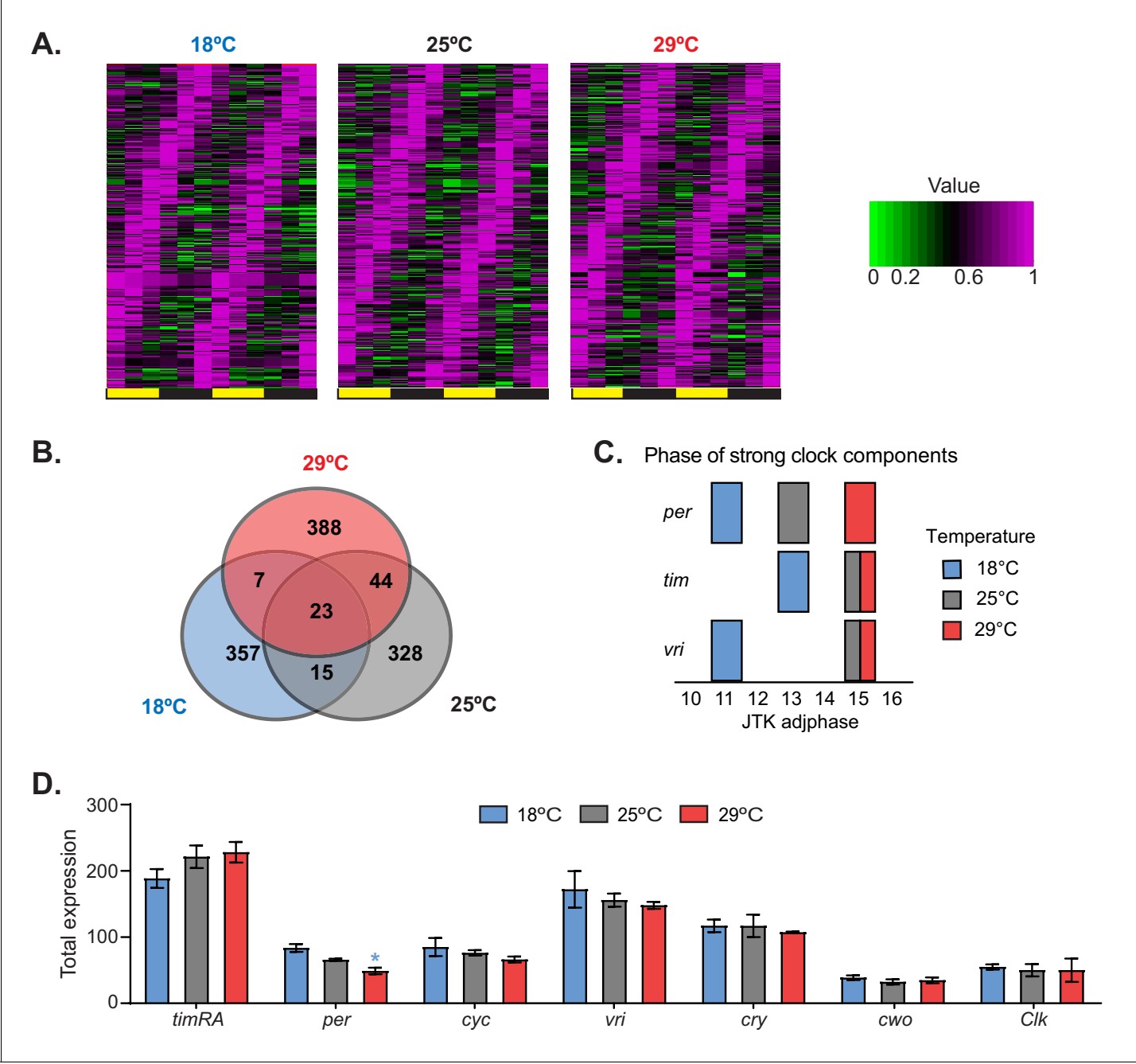

**Figure 1.** Temperature remodels the circadian transcriptome. (**A**) Heat maps of normalized expression of genes that display 24 hr cycling expression at different temperatures. Flies were entrained at the indicated temperatures for 3 days in 12:12 LD conditions. Fly heads were collected every 4 hr. Cycling expression was assessed as described in the 'Materials and methods' section. (**B**) Venn diagrams of the numbers of oscillating genes at different temperatures. (**C**) Phases of the core clock genes *tim*, *per*, and *vri* that display strong cycling at 18°C, 25°C, and 29°C. The blue, gray, and red blocks indicate the phases of these genes at 18°C, 25°C, and 29°C, respectively. (**D**) Total expression of the indicated clock genes in the microarray for each temperature. Only *per* shows a significant difference between 18°C and 29°C. Plotted are means ± SEM.

The online version of this article includes the following figure supplement(s) for figure 1:

**Figure supplement 1.** *tim* transcription is more efficient at 18°C than at 25°C or 29°C.
**Figure supplement 2.** Temperature regulates gene expression.

*1999*). Nevertheless, there were no major changes in the overall levels of most circadian components when gene expression at different temperatures was compared (*Figure 1D*).

To complement these data, we analyzed gene expression at the three temperatures utilizing oligonucleotide microarrays from samples that were collected at six time points and pooled before the assessment of gene expression. Pooling the samples allowed a more precise comparison of average gene expression. From the microarray analysis, we identified hundreds of mRNA transcripts that were expressed differentially as a function of temperature. When comparing to expression at 25℃, we observed both higher and lower abundance transcripts at both 18℃ and 29℃ (*Figure 1—figure supplement 2A*). A number of GO categories were enriched among the differentially expressed genes, and these categories spanned a wide range of biological functions (*Figure 1—figure supplement 2B*) consistent with previously reported data (*Boothroyd et al., 2007*). For example, we observed an increase in the expression of genes encoding ribosomal proteins, genes involved in protein folding, and genes involved in metabolic processes at 29℃ compared to that at 25℃. At 18℃, we saw increased expression of proteins involved in cuticle development compared to that at 25℃, which is consistent with previous reports showing that cuticle deposition is temperature dependent (*Boothroyd et al., 2007*; *Ito et al., 2011*). These results demonstrate the dramatic effect that temperature has on the steady-state levels of mRNAs involved in various cellular and metabolic processes, and specifically on the circadian molecular and transcriptional network.

## Temperature modulates *tim* alternative splicing

The 3′ RNA-seq datasets are a powerful tool for examining changes in gene expression. However, they lack the ability to capture many processing events, such as alternative splicing and alternative promoter usage. Therefore, we generated whole-transcript polyA$^+$ RNA-seq datasets from the heads of flies entrained at 18℃, 25℃, or 29℃. We found that temperature greatly impacted the pattern of alternative splicing of *tim* mRNA (*Figure 2A*). *per* is the only additional clock gene that showed some temperature-driven changes in alternative splicing, and the temperature-dependent alterations were not as striking as those observed in *tim*. However, the *per* splicing pattern has been shown to be changed drastically at temperatures lower than 18℃ (*Montelli et al., 2015*).

We identified four major isoforms that were generated from the *tim* locus (*Figure 2B*). The canonical transcript (referred to here as *tim-L*) did not significantly change in abundance with temperature, but the other three isoforms did. Two were more abundant at 18℃ than at 25℃. One of these isoforms was previously described as being induced by cold temperatures and is called *tim-cold* (*Boothroyd et al., 2007*; *Wijnen et al., 2006*; *Montelli et al., 2015*). *tim-cold* includes an additional intron near the 3′ end of the canonical *tim* mRNA. As a result, it has a longer 3′ UTR than *tim-L*, but a stop codon at the beginning of the intron results in a slightly smaller protein than that produced from the canonical isoform. The other cold-enriched isoform, although previously annotated, had not been characterized. We named this isoform *tim-short and cold* (*tim-sc*), as it putatively encodes a much smaller protein than does *tim-L*. *tim-sc* mRNA is generated by the usage of an alternative cleavage and polyadenylation site located within intron 10 of *tim*. By contrast, the other isoforms are also generated as the result of intron retention, but the ends of their 3′ UTRs are the same as that of the canonical isoform. The third isoform that we identified potentially encodes a protein that is smaller than those encoded by *tim-L* and *tim-cold* and larger than that encoded by *tim-sc*. This isoform was recently described and named *tim-tiny* (*Shakhmantsir et al., 2018*). However, to avoid confusion with *tim-sc* (which encodes an even smaller isoform), we will call it *tim-medium* (*tim-M*). This isoform is generated by the inclusion of an intron that contains a stop codon and shares all downstream exons with *tim-L*. Hence, it has the longest 3′ UTR of all *tim* isoforms. The isoform *tim-M* was present at higher abundance in flies entrained to 25℃ or 29℃ than in those kept at 18℃ (*Figure 2*).

As *tim-L, tim-cold, and tim-M* share the same 3′ end, they are indistinguishable in the 3′ RNA-seq datasets. Therefore, on the basis of the 3′ RNA-seq data, we were only able to conclude that the overall levels of these isoforms combined do not change with temperature (*Figure 2—figure supplement 1A*). By contrast, we detected a significant increase in the levels of *tim-sc* mRNA at 18℃ compared to those at 25℃ or 29℃ (*Figure 2—figure supplement 1B*). We confirmed the temperature-dependent differences by qRT-PCR using isoform-specific primers (*Figure 2—figure supplement 2A*). Unfortunately, the levels of *tim-L* could not be unequivocally

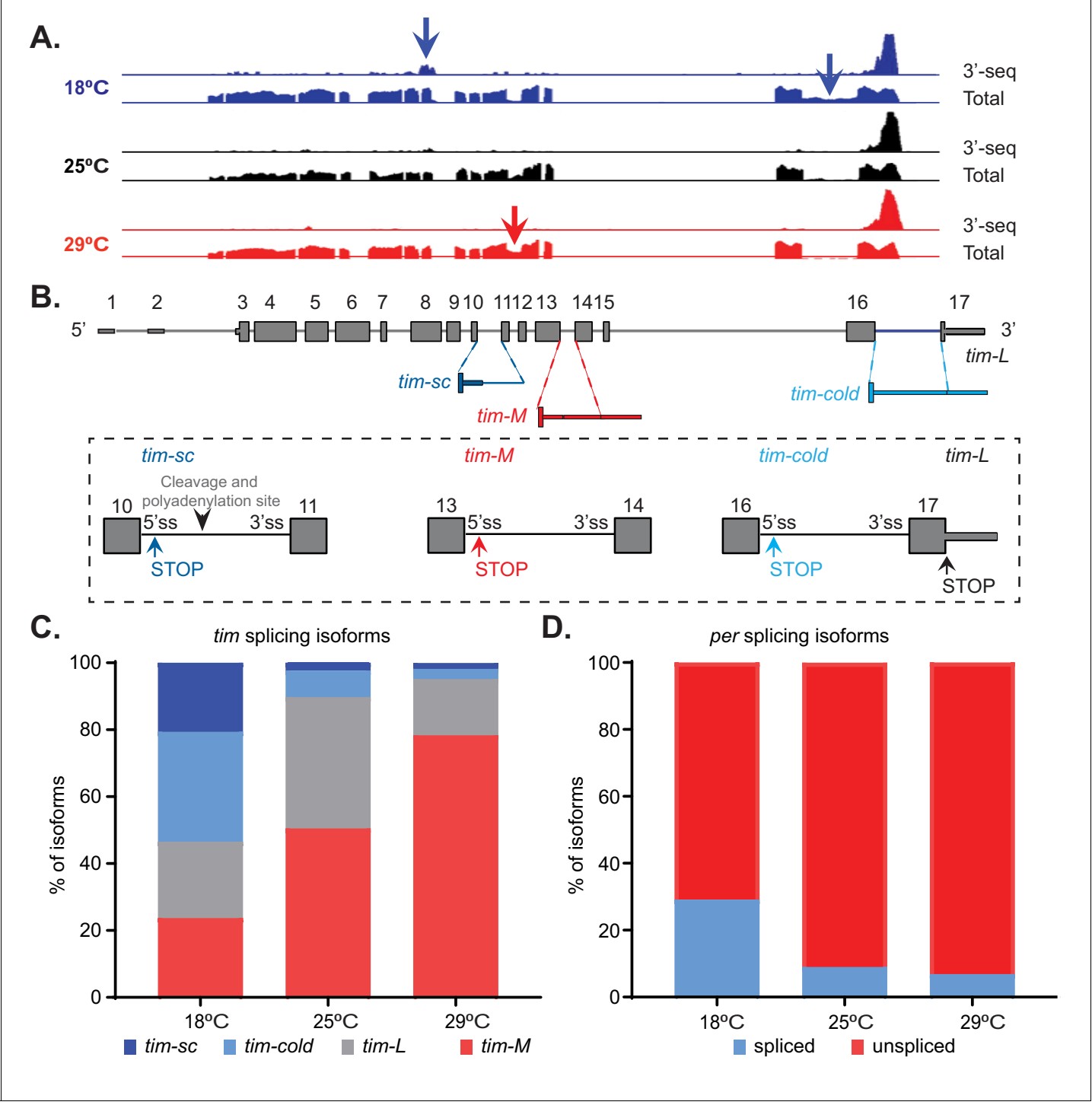

**Figure 2.** Temperature modulates *tim* alternative splicing. (**A**) Integrative Genomics Viewer (IGV) snapshot of the *tim* locus, indicating the expression of this gene at 18°C (blue), 25°C (black), and 29°C (red). The presented data include the aggregated data from the 3′ RNA-seq datasets presented in *Figure 1* (upper traces) as well as full transcript polyA$^+$ RNA-seq datasets (lower traces). The latter include two time points at 25°C and three time points at 18°C and 29°C. The arrows indicate the alternative splicing events that are regulated by temperature. (**B**) A schematic of the alternatively spliced *tim* isoforms. Constitutive exons are shown in gray, sequences found mainly at high temperatures in red, and sequences found mainly at low temperatures in blue. Close-ups of the exons surrounding each non-canonical isoform are shown in rectangles. Alternative stop codons (STOP) and cleavage and polyadenylation sites in these isoforms are also indicated. (**C**) Quantification of the relative amounts of *tim-sc* (dark blue), *tim-cold* (light blue), *tim-M* (red), and *tim-L* (gray) isoforms based on the data shown in panel (A). To quantify the different isoforms, we counted the relative number of

*Figure 2 continued on next page*

*Figure 2 continued*

spliced junctions that are specific for each isoform. (D) Quantification of the relative amount of spliced *per* (blue) and unspliced *per* (red) RNAs at the three assayed temperatures.

The online version of this article includes the following figure supplement(s) for figure 2:

**Figure supplement 1.** Relative expression of (A) *tim-L/M/cold* and (B) *tim-sc* in the two gene expression sets described in *Figure 1*.
**Figure supplement 2.** *tim* expression is regulated by temperature co- and post-transcriptionally.

assessed by qRT-PCR because its sequence overlaps with that of *tim-M.* These data revealed that the temperature-dependent changes in the splicing patterns of *tim* are at least as impressive as the previously described changes in the splicing of a small intron in the 3′ UTR of *per* (*Majercak et al., 1999*).

Using the whole-transcript RNA-seq datasets to quantify the relative amounts of each of the four isoforms (*Figure 2C*). We found that there was a dramatic change in the distribution of the four isoforms over the temperature range tested, from a fairly even distribution at 18°C to no expression of *tim-sc* and *tim-cold* and significant elevation of *tim-M* at 29°C. Surprisingly, we found that *tim-M* constitutes about 50% of *tim* mRNAs at 25°C, a temperature at which the level of *tim* expression is similar, if not higher, than that of the canonical isoform *tim-L.* This suggests the existence of previously unknown aspects of *tim* regulation, even in canonical conditions. In comparison to the temperature-dependent differences observed in *tim* alternative splicing, the well-characterized changes in *per* alternative splicing are very modest (*Low et al., 2008*) (*Figure 2D*).

To assess whether the observed changes in *tim* RNA processing that occur as a function of temperature also exist in the fly brain, we dissected the brains of flies entrained at 18°C, 25°C, and 29°C and measured the levels of each *tim* isoform by qRT-PCR. As observed when whole fly heads were analyzed, entrainment of the flies to lower temperatures (18°C) resulted in a strong increase of *tim-sc* and *tim-cold* levels compared to the levels in flies entrained at 25°C, and entrainment to higher temperatures (29°C) promoted *tim-M* expression (*Figure 2—figure supplement 2B*).

## Temperature-specific alternative splicing is conserved across *Drosophila* and correlates with temperature adaptation

If changes in the splicing pattern of *tim* are necessary for temperature adaptation, then these changes should be conserved among other *Drosophila* species that can adapt to temperature changes. Therefore, we analyzed the behavior and determined the splicing patterns of *tim* in three additional species belonging to the *Drosophila* genus entrained at 18°C, 25°C, and 29°C. Two of these species, *D. simulans* and *D. yakuba*, belong to the *Sophophora* subgenus, like *D. melanogaster*; the third, *D. virilis*, belongs to the *Drosophila* subgenus. *D. melanogaster* and *D. simulans* share the same tropical origin, but these species have become cosmopolitan, whereas *D. yakuba* is restricted to Africa (*Markow and O'Grady, 2007*; *Kuntz and Eisen, 2014*). *D. virilis*, a Holarctic species, is also geographically temperate. Although all three species displayed some degree of behavioral adaptation to temperature, we detected differences among them when analyzing the ratio between light and dark activity at the three assayed temperatures (*Figure 3A*). As expected, *D. melanogaster* had a temperature-dependent gradient in their light/dark activity ratio, with the highest ratios at 18°C (*Figure 3A*, left). For the *D. simulans* strain analyzed, we also observed a significant decrease in light/dark activity ratios as the temperature increased (*Figure 3A*, middle-left; *Figure 3—figure supplement 1A*, upper left panel), suggesting that *D. melanogaster* and *D. simulans* have similar behavioral adaptation to high and low temperatures. By contrast, for *D. yakuba*, the light/dark activity ratios at 25°C and 18°C were not different (*Figure 3A*, middle-right; *Figure 3—figure supplement 1A*, middle left panel). This suggests that this equatorial species has reduced capacity to adapt to lower temperatures when compared to the other species tested; this could be because this species does not experience large changes in temperature in the wild. It should be mentioned that although the light/dark activity ratio at 29°C for *D. virilis* was similar to that in the other species, its activity had the opposite trend: *D. virilis* was more active at night whereas the other species displayed reduced activity during the dark phase (*Figure 3A*, right; *Figure 3—figure supplement 1A*, lower left panel).

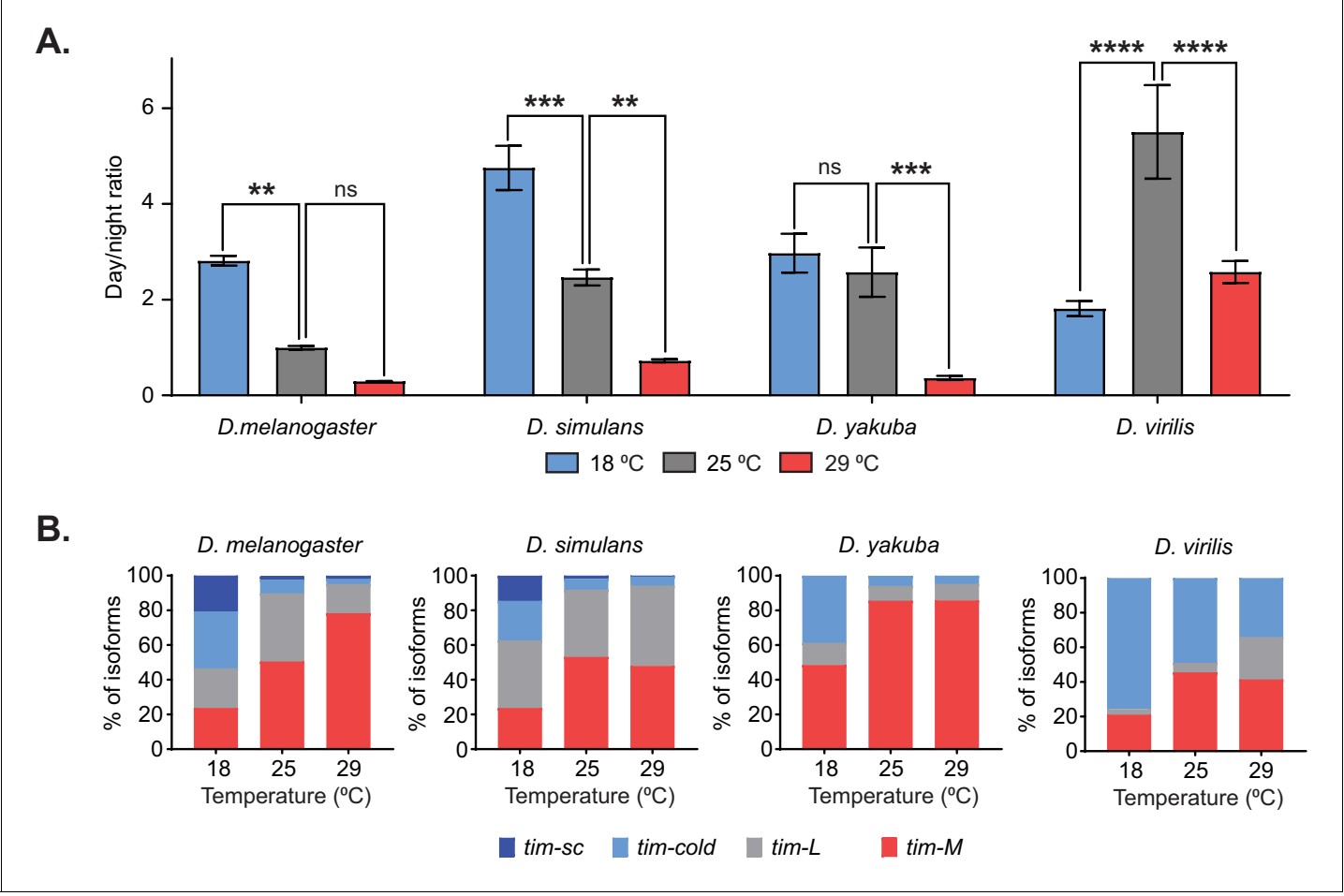

**Figure 3.** Temperature-specific alternative splicing is conserved across different *Drosophila* species. (**A**) Quantification of the ratio between day and night counts of locomotor activity for *D. melanogaster, D. simulans, D. yakuba*, and *D. virilis* at 18°C (blue), 25°C (gray), and 29°C (red). Each bar represents the ratio between the mean locomotor activity during the day and during the night over four days (n = 4) ± SEM. Significance was calculated by performing two-way ANOVA analysis (**, p<0.01; ***, p<0.001; ****, p<0.0001). (**B**) Quantification of the relative amounts of *tim-sc* (dark blue), *tim-cold* (light blue), *tim-M* (red), and *tim-L* (gray) from whole transcriptome polyA$^+$ mRNA-seq at the indicated temperatures. Each different isoform was quantified as in *Figure 2C*.

The online version of this article includes the following figure supplement(s) for figure 3:

**Figure supplement 1.** Temperature-specific alternative splicing is conserved across *Drosophila* species.

We next generated whole-transcript polyA$^+$ RNA-seq datasets from the heads of all three species entrained at the three different temperatures. All species displayed dynamic changes in *tim-cold* and *tim-M* levels in response to temperature changes, with *tim-cold* increasing and *tim-M* decreasing at lower temperatures (*Figure 3B*; *Figure 3—figure supplement 1A*, right). This suggests that these changes might be important for the behavioral changes shared by all species. However, we detected *tim-sc* only in *D. simulans* and *D. melanogaster*, which had very similar temperature-dependent alternative splicing profiles (*Figure 3B*; *Figure 3—figure supplement 1A*, right). As mentioned above, this isoform appears only at 18°C, suggesting that it might be relevant for the increased light/dark activity ratio observed for *D. simulans* and *D. melanogaster* when comparing the locomotor activity patterns at 18°C and 25°C. Surprisingly, high levels of *tim-cold* were detected in *D. virilis* at all of the assessed temperatures, including 29°C, suggesting that the mechanism is more complex in *D. virilis* than in the other species analyzed (*Figure 3B*). We also looked at changes in *per* 3′ UTR splicing in the different species as a function of temperature. *D. simulans* had a similar *per* splicing pattern to *D. melanogaster* (i.e., the intron is mostly retained and its splicing is partially increased in colder temperatures). On the other hand, *D. virilis* displayed no differences in *per*

splicing over this temperature range, whereas the changes in *D. yakuba* were slight (*Figure 3—figure supplement 1B*).

## Cold temperature decreases TIM-L levels by two independent mechanisms

As previously described (*Majercak et al., 1999*), we found that the levels of the canonical TIM-L protein were significantly downregulated at 18°C compared to those at 25°C or 29°C. We were unable to detect clearly or to differentiate the other isoforms by western blot (*Figure 4—figure supplement 1*). Nevertheless, *Boothroyd et al. (2007)* reported that TIM-COLD is expressed. *tim-M* mRNA is very abundant, but we did not detect a protein of the predicted size in the western blots, which strongly suggests that that this isoform is not translated or is translated very inefficiently. We reasoned that this isoform regulates in *cis* the amount of *tim-L* produced. This agrees with a recent publication that identified this mRNA isoform as non-functional and probably non-coding (*Shakhmantsir et al., 2018*). Although we observed bands of the size expected for TIM-SC in the western blot with anti-TIM antibody, these bands were present at all temperatures (*Figure 4—figure supplement 1*). We believe that these bands might represent canonical TIM degradation products that are of a similar size to TIM-SC. Moreover, the utilized TIM antibody was raised against the whole protein and detects TIM-SC with low efficiency, as shown by analysis of the protein fused to a FLAG tag described below.

Our sequencing data (*Figure 2—figure supplement 1A*) and reports by others (*Boothroyd et al., 2007*) indicate that the lower levels of TIM-L at 18°C compared to 25°C are not due to changes in total *tim* mRNA. This suggests that the lower TIM levels are the result of post-transcriptional and/or post-translational regulation. Recent work demonstrated that *tim* mRNA levels are regulated by miR-276 (*Chen and Rosbash, 2016*). To determine whether miRNA-mediated regulation of *tim* is temperature dependent, we tested the genome-wide association of mRNAs with AGO1 (the only miRNA-RISC effector protein in *Drosophila* [*Förstemann et al., 2007*]) at 18°C, 25°C, and 29°C. Briefly, we performed AGO1 immunoprecipitation from fly heads followed by hybridization to oligonucleotide microarrays. In these datasets, we can only assess overall *tim* levels as these microarrays cannot distinguish between *tim-L, tim-cold*, and *tim-M*.

We first confirmed that the overall distribution of AGO1-associated mRNAs (and hence miRNA-mediated regulation) is similar at all three temperatures (*Figure 4—figure supplement 2*). Interestingly, we observed temperature-dependent changes in the association of *tim* mRNA with AGO1 (*Figure 4A*). Binding of *tim* to AGO1 was very strong at 18°C, but there was little or no association at 25°C or 29°C. This pattern is *tim-L/M/cold*-specific: when we examined associations of mRNAs encoding other core circadian clock components with AGO1, we did not see such remarkable temperature-driven AGO1 association changes (*Figure 4A*). Some of the mRNAs encoding clock components did not bind at all (e.g., *cry, cyc*, and *per*), indicating that these transcripts are not regulated post-transcriptionally by miRNA at the tested temperatures. This was also the case for *tim-sc*, which did not bind to AGO1 at any temperature (*Figure 4A*). Others, bound to AGO1 at all temperatures (e.g., *Clk* and *vri*). There were small temperature-dependent changes in the association of *Clk* with AGO1, but it was highly bound to AGO1 at all temperatures.

To determine whether the association of the different *tim* transcripts to the miRNA-effector machinery is temperature dependent, we performed isoform-specific qPCRs from AGO1 immunoprecipitation samples. The mRNA encoding the transcription factor CBT (a known miRNA-regulated RNA; *Kadener et al., 2009*; *Lerner et al., 2015*) was strongly bound to AGO1 at all of the assayed temperatures. By contrast, we discovered that the association of *tim-L/M* (*tim-L* cannot be distinguished from *tim-M* as they share the 3′ UTR end), *tim-cold*, and *tim-M* to AGO1 was significantly higher at 18°C than at 25°C or 29°C (*Figure 4B*).

To determine the consequences of AGO1 binding on TIM expression, we generated flies carrying luciferase reporters fused to the different *tim* 3′ UTRs. All transgenes were inserted into the same genomic location, and we expressed these UAS-transgenes using the *tim-Gal4* driver. We assessed luciferase levels on these flies after three days of entrainment to 18°C or 25°C. We observed no significant temperature-dependent ($p < 0.53$) or interaction ($p < 0.46$) effects. However, the luciferase levels of the transgenes carrying the 3′ UTRs of *tim-L* and *tim-sc* were high compared to those of the other two transgenes, suggesting that they are under little (or no) post-transcriptional control. Interestingly, much less luciferase was expressed from the reporter carrying the 3′ UTR of *tim-cold* than

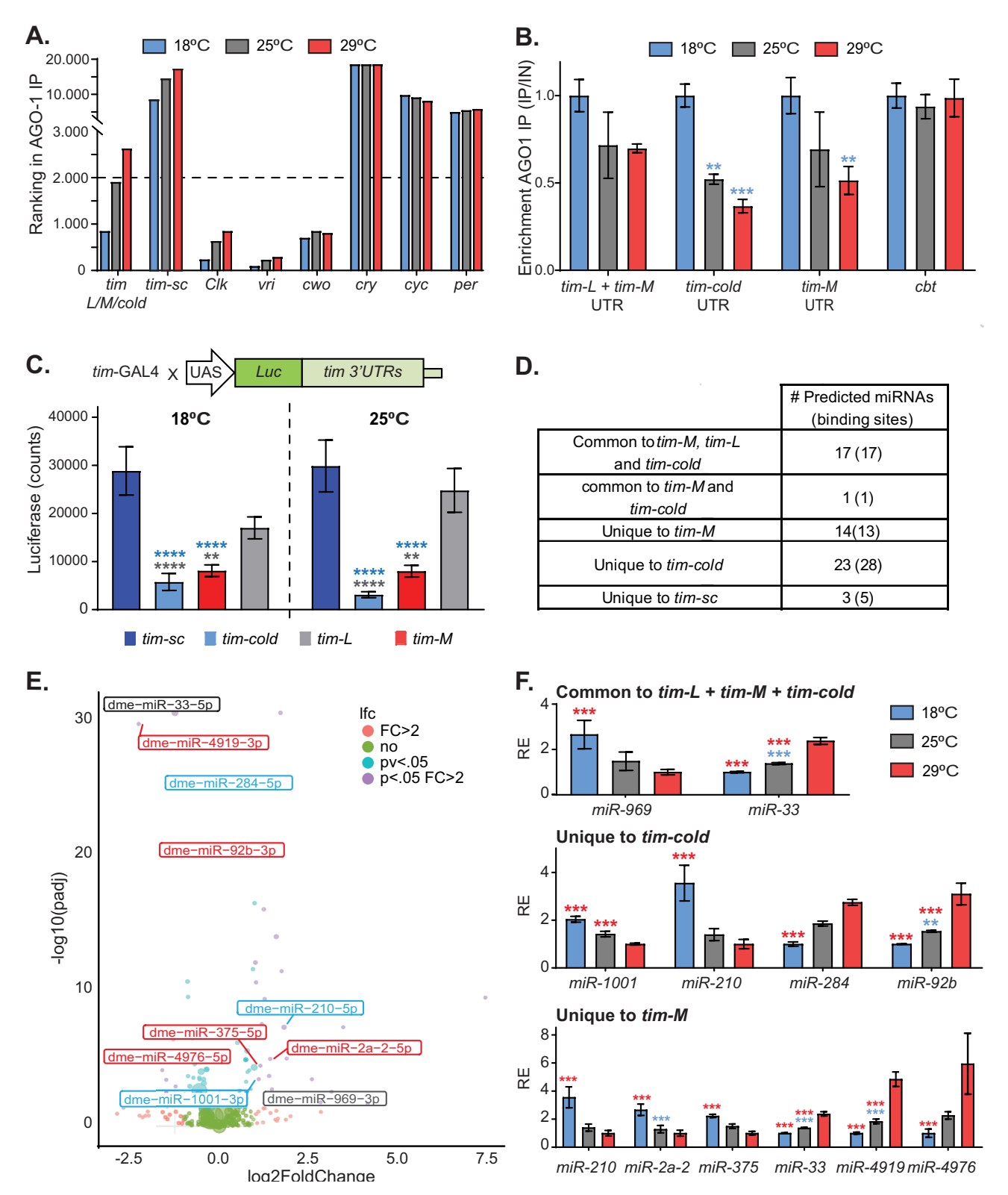

**Figure 4.** Alternatively spliced isoforms of *tim* are post-transcriptionally regulated. (**A**) Graph of the ranking of enrichment of RNAs in AGO-1 immunoprecipitates isolated from flies entrained at 18°C (blue), 25°C (gray), and 29°C (red). Briefly, we ranked RNAs according to their binding enrichment (lower ranking value corresponds to higher binding to AGO-1). To assess the ranking, we averaged the residuals in a linear model: signal in AGO1-immunoprecipitate/signal in input (n = 2 for each temperature). (**B**) *tim* isoform-specific qPCR of AGO1-bound mRNAs at 18°C (blue), 25°C (gray),

*Figure 4 continued*

and 29°C (red). Each bar represents an average of all time points normalized to a negative control (*cyc*). Blue stars indicate statistically significant levels relative to those at 18°C. Significance was determined by two-way ANOVA analysis using Tukey's multiple comparisons test (**, p<0.01; ***, p<0.001). Four different time points from each temperature were used as biological replica. Bars display the means ± SEM. (C) Luciferase assay of whole flies expressing *luciferase* fused to different *tim* 3′ UTRs at 18°C (left) and 25°C (right). Blue and gray stars represent statistically significant difference relative to *tim-sc* or *tim-L*, respectively, determined by performing two-way ANOVA analysis using Tukey's multiple comparisons test (*, p<0.05; **, p<0.01). Means ± SEM are plotted (n = 8). (D) Table summarizing the number of putative miRNA-binding sites identified in the different *tim* isoforms using TargetScan. (E) Volcano plot showing miRNAs that have differential expression at 18°C and 29°C (n = 4 for each temperature). The color code represents the significance for each miRNA differentiating all the combinations, for fold change >2 and adjusted p value <0.05. miRNAs with predicted binding sites present in any of the *tim* isoform 3′ UTRs were identified using TargetScanFly.6.2. Blue, gray, and red indicate putative targets for *tim-cold*, common sequence, or *tim-M* 3′ UTRs, respectively. (F) Relative abundances of miRNAs that are differentially expressed (fold change >2 and adjusted p value < 0.05) at different temperatures and that are predicted to target the 3′ UTR of the different isoforms. For each miRNA, the values were normalized to the minimum and their relative expression (RE) is plotted ± SEM. Blue and red stars represent statistical significance relative to expression at 18° or 29°C, respectively, determined using a negative binomial model employing DeSeq2. See details in the 'Materials and methods' (**, p<0.01; ***, p<0.001).

The online version of this article includes the following figure supplement(s) for figure 4:

**Figure supplement 1.** Western blot of TIM at 25°C (black) and 18°C (blue) using heads from *tim[01]* and *w[1118]* flies at ZT3 and ZT15.

**Figure supplement 2.** Density plot representing 4000 genes selected as 'AGO1-associated'.

from the other reporters, suggesting that the presence of this 3′ UTR strongly diminishes the translation of this isoform, by affecting either the stability or translation of the mRNA (*Figure 4C*). Although less luciferase was expressed from the *tim-M* 3′ UTR reporter than from the *tim-L* and *tim-sc* reporters, the difference was considerably less than that observed with the *tim-cold* 3′ UTR.

To further understand the post-transcriptional regulation of *tim*, we analyzed the different 3′ UTRs for potential miRNA-binding sites using TargetScan (*Agarwal et al., 2015*). The isoforms *tim-L*, *tim-cold*, and *tim-M* have a large number of putative miRNA-binding sites, many of which are evolutionary conserved and highly expressed in fly heads. *tim-sc* only had a handful of predicted miRNA-binding sites and was not bound to AGO-1 at any temperature, strongly suggesting that this isoform is not regulated by miRNAs (*Figure 4D*; *Supplementary file 3*). The miRNAs that are responsible for the association of *tim* isoforms with AGO1 might target only a temperature-specific isoform, might target more than one isoform, might be expressed in a temperature-dependent way, or might act through a combination of these mechanisms. We therefore sequenced AGO1-associated miRNAs from the heads of flies entrained to 12:12 LD cycles at 18°C, 25°C, and 29°C. Expression of most miRNAs was not affected by temperature (*Figure 4E*; *Supplementary file 4*). We first focused on the miRNAs targeting the 3′ UTR region common to *tim-L, tim-cold*, and *tim-M*. Of the 17 miRNAs that are predicted to target this region, the levels of only two differed with temperature. One of them (miR-33) was increased at 29°C in comparison with the level at 18°C. Notably miR-969, which contains what is predicted to be a strong binding site in the 3′ UTR region common to *tim-L, tim-cold*, and *tim-M*, was strongly (~3-fold) upregulated at 18°C compared to 29°C (*Figure 4F*). However, *tim-M* and *tim-cold* contain additional 3′ UTR regions, and these regions are predicted to be targeted by other miRNAs that are strongly regulated by temperature. Half of the miRNAs predicted to bind these *tim* isoforms were highly expressed at 18°C and half were highly expressed at 29°C (*Figure 4F*; *Supplementary file 3* and *4*). This strongly suggests the existence of a miRNA-mediated mechanism that downregulates the expression of TIM-COLD and TIM-M at all temperatures. That TIM-COLD is indeed present at 18°C (*Boothroyd et al., 2007*; *Wijnen et al., 2006*) suggests that the main function of the miRNA-mediated regulation of this isoform is to impose a threshold that blocks the expression of this protein at temperatures other than 18°C, when the transcription levels of this isoform are lower.

In summary, these results strongly suggest that at 18°C, differences in the expression of the canonical TIM protein relative to levels at 25°C are due to the deviation of *tim* transcription toward the production of *tim-sc* and *tim-cold*, which is under strong post-transcriptional regulation. In addition, *tim-L, tim-M*, and *tim-cold* are subjected to stronger post-transcriptional control by miRNAs than are the other isoforms.

## Overexpression of TIM isoforms have different effects on circadian behavior

We next sought to determine the functionality of the different TIM isoforms. To do so, we generated plasmids to express each TIM protein isoform fused to a C-terminal FLAG tag; all coding regions were fused to the same 3′ UTR and were under the control of the UAS-promoter (*Figure 5—figure supplement 1A*). To verify the expression of TIM from these plasmids, we co-transfected them into *Drosophila* S2 cells (which do not express *tim*) together with a GAL4 expression plasmid. As expected, using an anti-FLAG antibody, we observed the bands of predicted sizes for TIM-SC, TIM-M, TIM-COLD, and TIM-L (*Figure 5—figure supplement 1B*, left). Interestingly, anti-TIM antibodies efficiently detected TIM-L and TIM-COLD but not TIM-SC (*Figure 5—figure supplement 1B*, right). Moreover, we observed a degradation product that overlapped with TIM-SC, demonstrating that it is very difficult (if not impossible) to detect this isoform unequivocally by western blot.

We next generated flies that expressed these TIM proteins under control of the *tim-Gal4* driver by targeting the plasmids to the *attB* insertion site. We determined the locomotor activity patterns of these flies at 25˚C. When kept under a 12:12 LD cycle, we observed a clear advance in the start of the evening activity component in flies that overexpressed *tim-sc* compared to *tim-gal4* controls, but no changes in the morning anticipation (*Figure 5A–C*). This, together with the fact that in DD over-expression of TIM-SC led to more than 1 hr shortening of the period compared to the control (*Figure 5D*), suggests a role for *tim-sc* mRNA and protein in the behavioral advance observed in wild-type flies at 18˚C.

In agreement with the results described by *Yang and Sehgal (2001)*, overexpression of TIM-L at 25˚C resulted in a high percentage of flies with long, weak, or no rhythms in DD conditions (*Figure 5D,E*). Overexpression of TIM-COLD also resulted in slightly weaker and/or longer rhythms, although to a lesser degree than is observed upon TIM-L overexpression. This suggests that the two proteins are not fully equivalent, even though both are functional. Surprisingly, overexpression of TIM-M did not significantly change the locomotor activity pattern in either LD or DD, suggesting that this protein is non-functional (*Figure 5D,E*). When we performed the behavioral experiments at 18˚C, we observed similar results: shortening of the period when TIM-SC was overexpressed and weaker rhythmicity when TIM-L was overexpressed (*Figure 5D*).

## TIM-SC rescues the behavioral rhythms of tim[01] flies in LD and DD

To test whether TIM-SC is functional, we determined whether overexpression of this protein is suffi-cient to rescue the behavioral rhythms of *tim[01]* flies. We generated flies that overexpressed the FLAG-tagged TIM-SC or TIM-L protein in a *tim[01]* background and overexpressed them using the *tim-Gal4* driver. As previously reported (*Ousley et al., 1998*), TIM-L partially rescued the behavioral rhythms of *tim[01]* mutants. The rescue was not total, probably because of the high level of expression of the UAS-TIM transgene. Interestingly, we found that TIM-SC also rescued the rhythms of *tim[01]* flies both in LD and in DD. In LD, expression of the *tim-sc* RNA resulted in the rescue of the behav-ioral rhythms (evening anticipation) in more than 60% of the flies (*Figure 6A–C*). Moreover, more than 40% of flies displayed rhythmic behavior in DD. Most of these rescued flies have a shorter period than the flies rescued with the canonical isoform (*Figure 6B,C*). Similar, but dampened, results were obtained when performing the experiment at 18˚C: expression of *tim-L* or *tim-sc* in *tim[01]* flies rescued the evening advance in 35% of the flies in LD. In addition, 20% of the flies expressing *tim-L* and 30% of the flies expressing *tim-sc* displayed rhythmic behavior in DD condi-tions. These results demonstrate that, despite its smaller size, the protein produced from *tim-sc* is functional.

## TIM-SC binds to but does not stabilize PER

TIM-L binds to and stabilizes PER, which is key for proper repression of CLK-driven transcription (*Nawathean and Rosbash, 2004*). The interaction between TIM and PER happens through two dif-ferent regions in TIM: one that includes the nuclear localization signal and another located within the region from amino acid 715 through amino acid 914 (*Saez and Young, 1996*). *tim-sc* contains only the first 892 amino acids of the canonical TIM. Hence, we tested whether TIM-SC was able to bind to and stabilize PER in *Drosophila* S2 cells, which do not naturally express either TIM or PER. We co-transfected these cells with a PER expression plasmid with or without a plasmid expressing each TIM

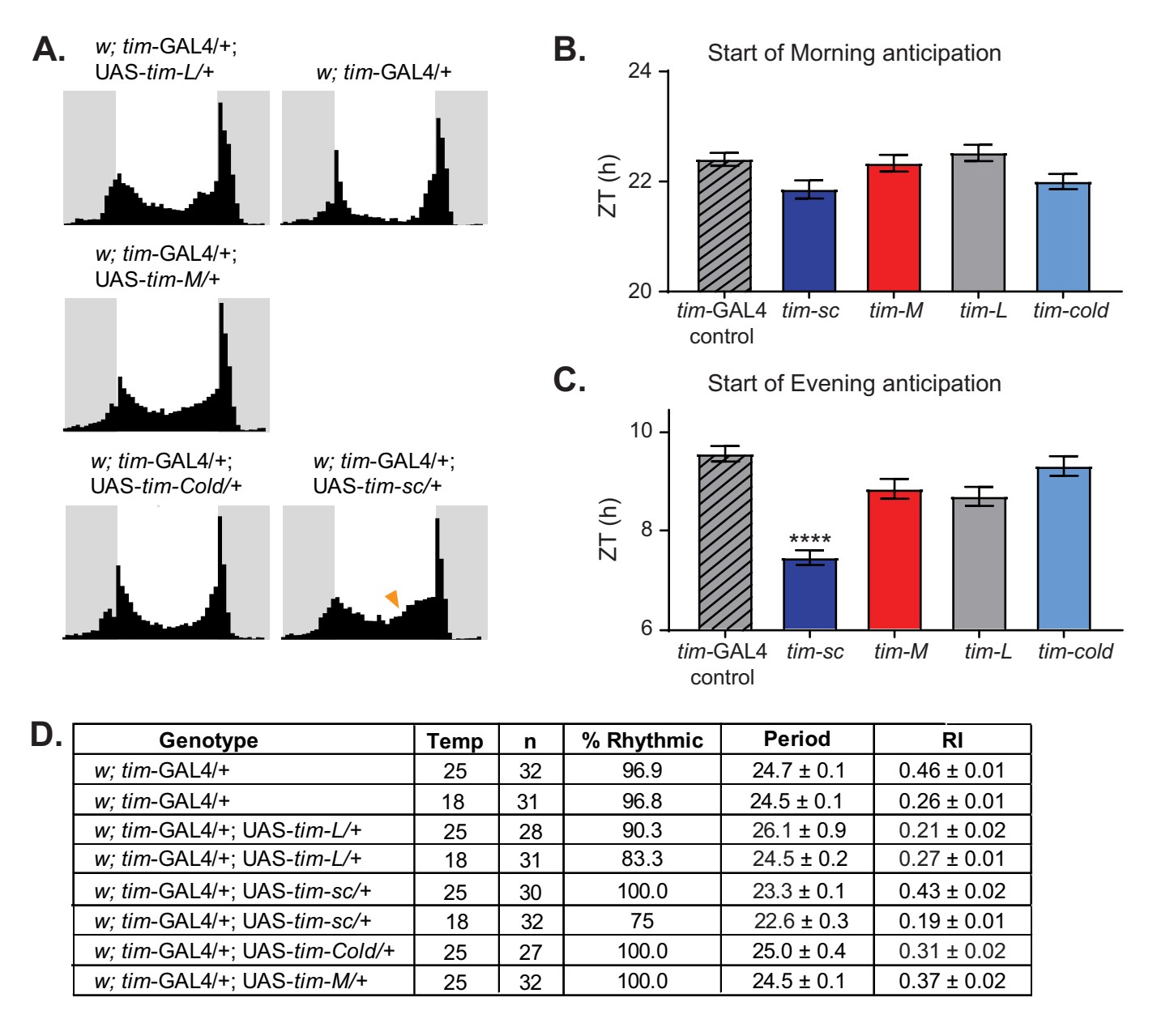

**Figure 5.** Overexpression of *tim* isoforms at 25°C results in various behavioral defects. (**A**) Average activity over three days in 12:12 LD at 25°C of flies of the indicated genotypes. Light phase is represented in white and dark-phase in gray. (**B**) Quantification of the start of the morning activity component in the last three days of LD (n = 27–31) ± SEM. (**C**) Quantification of the start of the evening activity component in the last three days of LD (n = 27–31) ± SEM. Stars represent statistical significance relative to *tim*-Gal4 control calculated by one-way ANOVA using Tukey's multiple comparisons test (****, p<0.0001). (**D**) Table summarizing the behavior of the indicated genotypes in DD at 18°C or 25°C, indicating percent rhythmic, period length, rhythmicity index (RI), and respective SEMs.

The online version of this article includes the following figure supplement(s) for figure 5:

**Figure supplement 1.** Overexpression of different *tim* isoforms with a FLAG-tagged C-terminus.

isoform. As previously described (*Nawathean and Rosbash, 2004*), TIM-L strongly stabilized PER (*Figure 6D*, top). However, the expression of TIM-SC did not result in PER stabilization (*Figure 6D*, top). This was not a consequence of the levels and/or stability of this TIM isoform, as we observed even higher amounts of TIM-SC than of TIM-L in the cells upon transfection (*Figure 6D*, bottom). To determine whether TIM-SC bound to PER, we performed co-immunoprecipitation experiments in

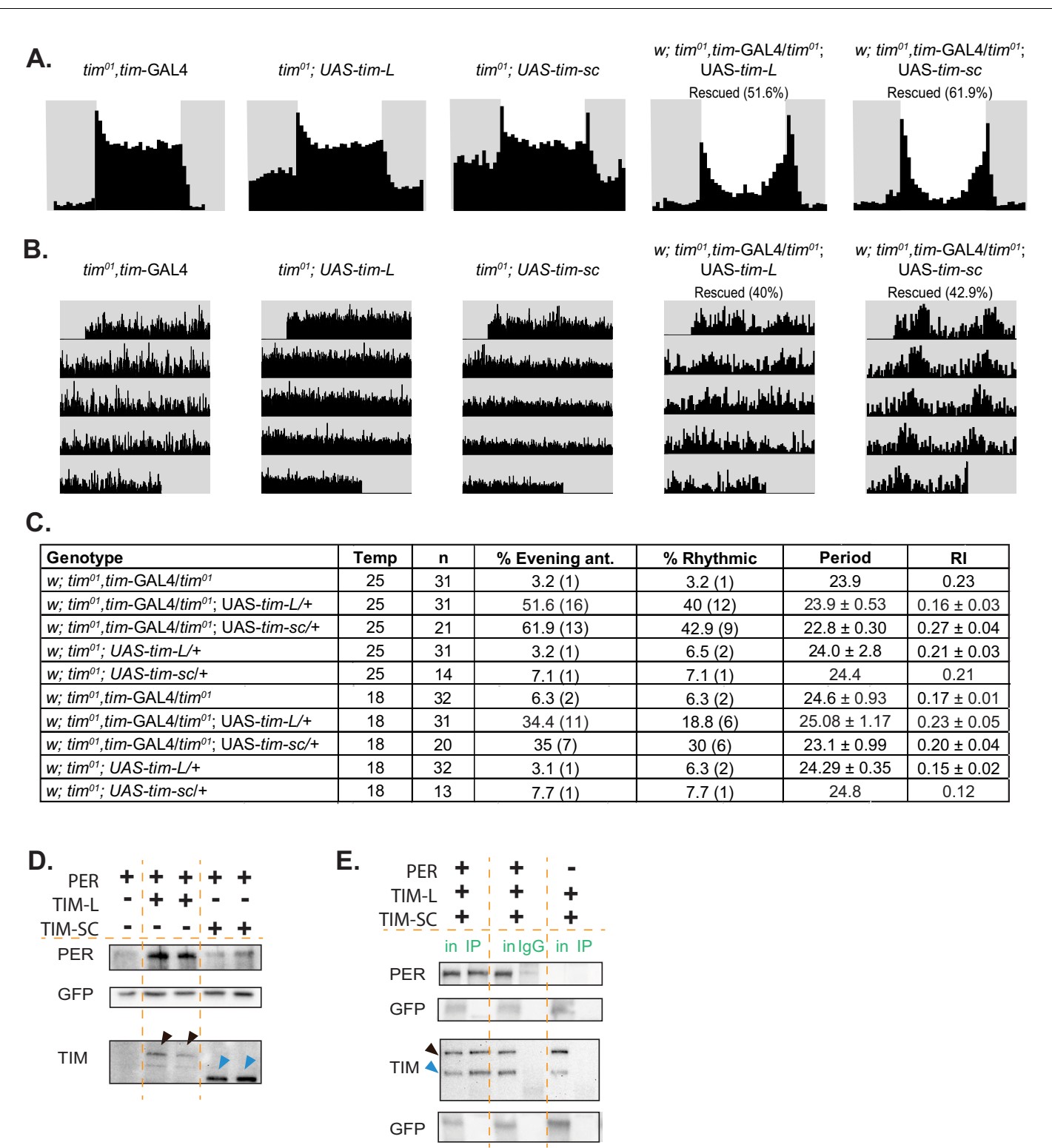

| Genotype | Temp | n | % Evening ant. | % Rhythmic | Period | RI |
|---|---|---|---|---|---|---|
| *w; tim$^{01}$,tim*-GAL4/*tim$^{01}$* | 25 | 31 | 3.2 (1) | 3.2 (1) | 23.9 | 0.23 |
| *w; tim$^{01}$,tim*-GAL4/*tim$^{01}$*; UAS-*tim-L*/+ | 25 | 31 | 51.6 (16) | 40 (12) | 23.9 ± 0.53 | 0.16 ± 0.03 |
| *w; tim$^{01}$,tim*-GAL4/*tim$^{01}$*; UAS-*tim-sc*/+ | 25 | 21 | 61.9 (13) | 42.9 (9) | 22.8 ± 0.30 | 0.27 ± 0.04 |
| *w; tim$^{01}$*; UAS-*tim-L*/+ | 25 | 31 | 3.2 (1) | 6.5 (2) | 24.0 ± 2.8 | 0.21 ± 0.03 |
| *w; tim$^{01}$*; UAS-*tim-sc*/+ | 25 | 14 | 7.1 (1) | 7.1 (1) | 24.4 | 0.21 |
| *w; tim$^{01}$,tim*-GAL4/*tim$^{01}$* | 18 | 32 | 6.3 (2) | 6.3 (2) | 24.6 ± 0.93 | 0.17 ± 0.01 |
| *w; tim$^{01}$,tim*-GAL4/*tim$^{01}$*; UAS-*tim-L*/+ | 18 | 31 | 34.4 (11) | 18.8 (6) | 25.08 ± 1.17 | 0.23 ± 0.05 |
| *w; tim$^{01}$,tim*-GAL4/*tim$^{01}$*; UAS-*tim-sc*/+ | 18 | 20 | 35 (7) | 30 (6) | 23.1 ± 0.99 | 0.20 ± 0.04 |
| *w; tim$^{01}$*; UAS-*tim-L*/+ | 18 | 32 | 3.1 (1) | 6.3 (2) | 24.29 ± 0.35 | 0.15 ± 0.02 |
| *w; tim$^{01}$*; UAS-*tim-sc*/+ | 18 | 13 | 7.7 (1) | 7.7 (1) | 24.8 | 0.12 |

**Figure 6.** Overexpression of *tim-sc* at 25°C partially rescues *tim$^{01}$* flies in LD and DD, and TIM-SC binds to PER. (**A**) Average activity over three days in 12:12 LD at 25°C of flies of the indicated genotypes. The light phase is represented in white and the dark-phase in gray. The percentage of rescued flies (displaying evening anticipation) is reported in brackets. (**B**) Actograms of each genotype in DD at 25°C. (**C**) Summary of the percentage of rescue in LD (evening anticipation component) and in DD (percent rhythmic, period length, rhythmicity index, and their respective SEMs). (**D**) Representative western blot showing staining for PER (top), GFP (middle), and TIM (bottom) of protein lysates from S2 cells transfected with pAcPerV5 (PER), pAcTimL-HA (TIM-

*Figure 6 continued on next page*

Figure 6 continued

L), and pAcTimSC-HA (TIM-SC); pAcGFP was used as a transfection control (n = 3). Arrows in the TIM staining point to TIM-L (black) and TIM-SC (blue). (E) Representative western blot for PER, GFP, and TIM of PER immunoprecipitates in S2 cells co-transfected with pAcPerV5 (PER), pAcTimL-HA (TIM-L), or pAcTimSC-HA (TIM-SC) (n = 2). IgG and no PER controls were loaded in the middle and right of the membrane. pAcGFP was used as a transfection control. Arrows indicate TIM-L (black) and TIM-SC (blue). in, input; IP, immunoprecipitation.

The online version of this article includes the following figure supplement(s) for figure 6:

**Figure supplement 1.** TIM-SC binds to PER.

**Figure supplement 2.** Overexpression of TIM-SC in *per01* results in a peak of activity resembling the evening activity component in a small fraction of the flies.

cells transfected with PER and TIM-SC and/or TIM-L. Interestingly, we found that both TIM-L and TIM-SC bound to PER (*Figure 6E*; *Figure 6—figure supplement 1*). Moreover, TIM-SC bound to PER even in presence of TIM-L (*Figure 6E*). These results suggest that TIM-SC might function independently of PER.

Therefore, we next determined whether overexpression of TIM-SC alters the locomotor activity of *per01* flies. As expected, *per01* cannot anticipate either the light-on or the light-off transition (*Figure 6—figure supplement 2A*). Overexpression of TIM-L did not alter the behavioral patterns of these flies. Interestingly, we observed that a small fraction (~19%) of the *per01* flies overexpressing TIM-SC did indeed show a peak of activity resembling the evening activity component (*Figure 6—figure supplement 2A*), suggesting that TIM-SC could alter behavior in a PER-independent way, at least in some conditions. Importantly, we did not observe any type of rescue of the behavioral rhythmicity of *per01* flies by overexpression of TIM-SC in DD conditions (*Figure 6—figure supplement 2B*). This was not surprising because PER is essential for circadian rhythmicity. From these results, we concluded that TIM-SC and TIM-L act by two different mechanisms and that TIM-SC is less dependent on PER function.

## Elimination of tim-sc results in changes in tim processing and locomotor activity

To further understand the role of *tim-sc*, we generated flies in which the cleavage and polyadenylation site used for the generation of *tim-sc* mRNA was mutated using the CRISPR method (40A flies, *Figure 7A*). As expected, these flies do not express *tim-sc* at any temperature as shown by qRT-PCR (*Figure 7B*). Isogenic control flies had an increased day/night activity ratio after transfer to 18°C in comparison to the flies maintained at 25°C and an advance of the evening behavioral component (*Figure 7C–E*). Interestingly, 40A flies had cold-like activity patterns of activity even at 25°C. For example, at 18°C and 25°C, 40A mutants display lower activity than control flies only at night. At 29°C, the 40A flies were less active in both the light and dark periods, partly because of a very distinctive midday siesta behavior (*Figure 7D*; *Figure 7—figure supplement 1A,B*). Even more importantly, and opposite to TIM-SC overexpression, 40A mutants displayed a significant delay in the time of evening activity onset that was greater at 18°C but also significant at 25°C (*Figure 7C and E*).

To determine the molecular consequences of the disruption of *tim-sc* production, we measured the levels of the other *tim* isoforms in control and 40A flies at 18°C, 25°C, and 29°C. The *tim-sc* mutant flies had levels of *tim* mRNA comparable to levels in control flies as assessed by determining the levels of the constitutive exons 5 and 6 (*Figure 7F*, left). However, elimination of *tim-sc* resulted in an increase in the levels of *tim-cold* both at 18°C and at 25°C at the maximum expression time point (ZT15; *Figure 7E*). *tim-L/M* levels were also increased in the mutant but only at 25°C. These results strongly suggest that both *tim-sc* production and TIM-SC protein regulate the daily pattern of locomotor activity and the response to temperature changes. Importantly, we observed some degree of temperature adaptation in 40A mutants, which we postulate is mediated by the increased production of *tim-cold*. tim alternative splicing is regulated directly by temperature in vitro and in vivo.

To obtain insight into the mechanism by which temperature regulates *tim* alternative splicing, we analyzed the temperature dependence of *tim* alternative splicing in *tim01* and *per01* mutants. We entrained control and mutant lines in LD at 18°C and at 29°C, collected flies every 4 hr, extracted RNA from fly heads from a mix of all time points, and performed qRT-PCR. We observed a similar

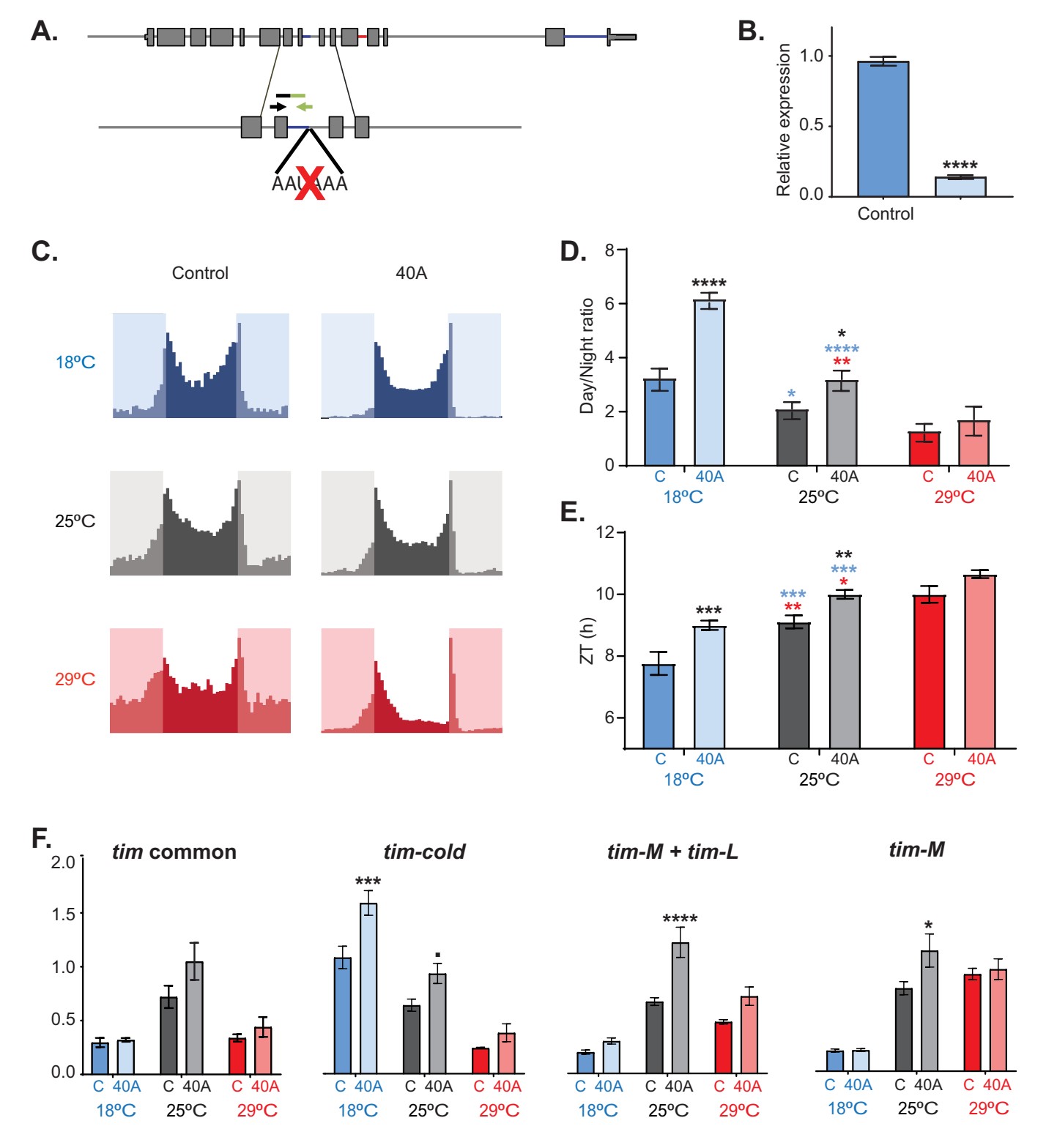

**Figure 7.** *tim-sc* null flies have behavioral and molecular defects. (**A**) Schematic of the mutation performed by CRISPR used to generate *tim-sc* null flies (40A flies). The black and green arrows indicate the locations of the forward and reverse primers used for the verification of *tim-sc* depletion. (**B**) Levels of *tim-sc* in 40A flies. The displayed qPCR result was obtained using RNA obtained from the heads of flies entrained at 18˚C (n = 5). (**C**) Average activity during three days of control (left) and 40A (right) flies at 18˚C (blue), 25˚C (gray), and 29˚C (red) in 12:12 LD cycles (n = 23–32). (**D**) Ratio between total light/dark activity at 18˚C (blue), 25˚C (gray), and 29˚C (red) in 12:12 LD. Blue and red stars represent statistically significant difference between flies kept

*Figure 7 continued on next page*

*Figure 7 continued*

at 18°C or 29°C and those at 25°C; black stars indicate differences relative to the control flies. Significance was determined by two-way ANOVA using Sidak's multiple comparisons test (n = 16–31). (E) Quantification of the start of the evening activity component on day 3 of 12:12 LD. Blue and red stars represent statistically significant differences between flies kept at 18°C or 29°C and those kept at 25°C, whereas black stars represent differences relative to the control flies, as determined by two-way ANOVA using Sidak's multiple comparisons test (n = 16–31). (F) Assessment by qRT-PCR of exons 5 and 6 (common to all *tim* isoforms, left), *tim-cold* (middle left), *tim-L* + *tim-M* (middle right), and *tim-M* (right) at 18°C (blue), 25°C (gray), and 29°C (red) at ZT15 in 12:12 LD. Significance was determined by two-way ANOVA using Sidak's multiple comparisons test for each isoform (n = 5). Each bar represents the mean ± SEM. *, p<0.05; **, p<0.01; ***, p<0.001; ****, p<0.0001.

The online version of this article includes the following figure supplement(s) for figure 7:

**Figure supplement 1.** Mean total activity counts in control and 40A mutant flies at 18°C (blue), 25°C (gray), and 29°C (red) in 12:12 LD during the day (A) and night (B) ± SEM (n = 23–32).

trend in all three fly lines: there were higher levels of *tim-cold* and *tim-sc* and lower levels of *tim-M* at 18°C compared to 29°C (*Figure 8A*). This demonstrated that, as previously shown for *per* temperature-dependent splicing (*Collins et al., 2004*), thermosensitive *tim* splicing events are independent of the circadian clock.

The splice sites flanking alternatively spliced exons are usually weaker than those flanking constitutively spliced exons, and temperature is known to influence the efficiency of splicing strongly (*Jakšić and Schlötterer, 2016*). Hence, we determined the strengths of the different splice sites in *tim* using publicly available software (*Reese et al., 1997*; *Low et al., 2008*). Almost all *tim* constitutive introns had strong 5′ and 3′ splice sites (scores above 0.6, *Figure 8—figure supplement 1*). Interestingly, the introns retained in *tim-M* and in *tim-cold* have a strong 5′ splice site and a weak 3′ splice site. This suggests that these isoforms result from alternative splicing due to the presence of weak 3′ splice sites. However, the splice sites of the intron associated with *tim-sc* have strong 5′ and 3′ splice sites (*Figure 8—figure supplement 1*). In addition, we observed that in *D. simulans* and *D. yakuba*, the introns retained in *tim-cold* and *tim-M* have strong 5′ splice sites and weak 3′ splice sites; alternative splicing is temperature-sensitive in these species (*Figure 8—figure supplement 1*).

*tim* was one of the few genes in which we observed temperature-dependent changes in alternative splicing (data not shown). Hence, it is possible that *tim* alternative splicing itself could be thermosensitive. To test this possibility, we utilized the *Drosophila* S2 cells that do not express most circadian components, including *Clk*, *per* and *tim*. However, *tim* expression can be induced by expression of CLK (*McDonald and Rosbash, 2001*). We transfected these cells with a plasmid that drives the expression of *Clk* under the control of the *metallothionein* conditional promoter. After the induction of *Clk*, we cultured the cells at either 25°C or 18°C for 24 hr. We then collected the cells, extracted RNA, and characterized the splicing pattern of *tim* by qRT-PCR. For each of the alternative isoforms, we measured the ratio between the unspliced and spliced variants and compared the ratios obtained for the cells incubated at the two temperatures. This experiment reproduced the results obtained in vivo: the levels of *tim-cold* and *tim-sc* were higher at 18°C than at at 25°C, whereas *tim-M* expression showed the opposite trend (*Figure 8B*). We did not observe temperature-induced changes in splicing of a constitutive *tim* exon (Control, *Figure 8B*), demonstrating that this effect is specific for the thermosensitive introns.

One intriguing possibility is that *tim* splicing itself functions as a thermometer. This could be accomplished if a temperature change, for example, results in the exposure of binding sites for a specific splicing factor. To test the possibility that splicing serves as a temperature sensor, we generated three different *tim* minigene reporters, consisting of the exons and intron involved in each *tim* alternative splicing event (*Figure 8C*). As observed in vivo and in the context of the whole *tim* gene in S2 cells, we observed a temperature-dependent effect on the splicing of the three different *tim* minigenes. We did not observe any temperature dependence in the splicing of a minigene generated from a constitutive exon (*Figure 8D*), demonstrating a direct and specific effect of temperature on *tim* alternative splicing events.

To further understand the mechanism by which the different introns sense temperature, we used the algorithm Mfold (*Zuker, 2003*) to model the structure of the temperature-sensitive introns and their flanking exons. Interestingly, we found that the 5′ splice sites of the introns retained at low temperatures (i.e., those included in *tim-sc* and *tim-cold*) are predicted to be

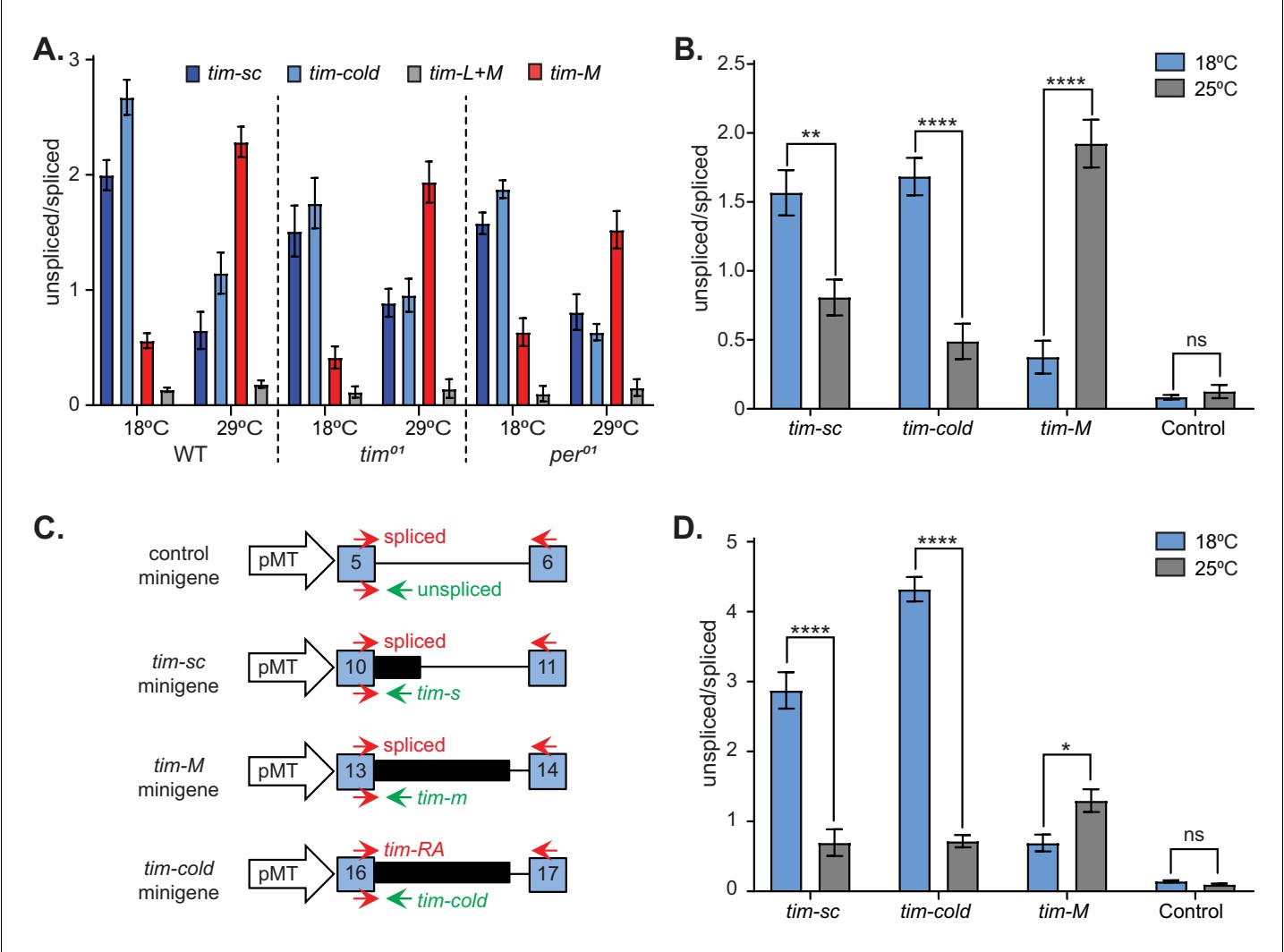

**Figure 8.** Splicing of *timeless* is temperature dependent and clock independent. (A) qRT-PCR results from wild-type, *tim$^{01}$*, and *per$^{01}$* clock mutant flies entrained at 18˚C or at 29˚C. Each bar represents the mean ratio (± SEM) between the unspliced *tim* and *tim-sc* (dark blue), *tim-cold* (light blue), *tim-M* (red), and *tim* control (gray). (B) *Drosophila* S2 cells transfected with pMT-*Clk* display the same *tim* alternative splicing patterns observed in living flies at 18˚C (blue) and at 25˚C (gray). Each bar represents the mean ratio (± SEM) between the unspliced and spliced variant of *tim-sc, tim-cold, tim-M* and *tim* control. Significance was determined by two-way ANOVA using Sidak's multiple comparisons test (n = 3). (C) Schematic of the *tim* isoform minigenes and locations of the primers used to measure the spliced (top) and unspliced (bottom) isoforms of each intron. (D) Mean unspliced/spliced ratios (± SEM) for the indicated minigenes at 18˚C (blue) and at 25˚C (gray). Significance was determined by two-way ANOVA using Sidak's multiple comparisons test (n = 3). *, p<0.05; **, p<0.01; ***, p<0.001.

The online version of this article includes the following figure supplement(s) for figure 8:

**Figure supplement 1.** Table summarizing the 3′ and 5′ splicing strengths of each *tim* intron for *D. melanogaster, D. simulans,* and *D. yakuba.*

**Figure supplement 2.** Predicted RNA secondary structures for (A) *tim-sc,* (B) *tim-cold,* (C) the *tim-sc* mutant without the cleavage and polyadenylation site, and (D) *tim-M* at 25˚C for each exon-intron-exon region.

engaged in strong secondary structures (*Figure 8—figure supplement 2A and B*). As RNA structures are more stable at lower temperatures, we postulate that these 5′ splice sites are less accessible at 18˚C than at 25˚C or 29˚C, rendering these introns temperature-sensitive. There were no differences in the predicted secondary structures of the 5′ splice sites when the cleavage and polyadenylation sites were removed (*Figure 8—figure supplement 2C*). By contrast, the 5′ splice site of the intron retained at high temperature (which is part of *tim-M*) was not predicted to be part of a stable secondary structure (*Figure 8—figure supplement 2D*). The

decreased splicing efficiency of the *tim-M* intron at 29°C could be a consequence of the lower general efficiency of splicing at higher temperatures.

## Discussion

In this study, we show that temperature dramatically and specifically changes the splicing pattern of the core circadian component *tim*. We found that the lower levels of the canonical TIM protein, TIM-L, that occur at 18°C compared to 25°C are due to the induction of two cold-specific splicing isoforms, *tim-cold* and *tim-sc*. The isoform *tim-cold* encodes a protein that is very similar to TIM-L, but its expression is under strong post-transcriptional control, as shown by the analysis of transcripts associated with AGO1 and in vivo luciferase reporter assays. Interestingly, temperature-dependent alterations in *tim* splicing patterns are conserved in several *Drosophila* species and correlate well with the capacity of the species to adapt their activity to temperature changes in the analyzed strains. However, we noticed that the behavioral pattern of *D. virilis* at the tested temperatures is intriguing, and it would be interesting to determine whether other strains of this species show the same patterns of behavior and splicing.

We found that overexpression of TIM-SC advances the phase and shortens the period of the circadian clock, suggesting that the expression of this isoform might mediate some of the changes seen upon induction of flies at 18°C. In addition, expression of TIM-SC rescues $tim^{01}$ phenotypes in LD (~50% of the flies that express TIM-SC display evening anticipation) and in DD (a similar number of flies show rhythmic behavior, in most flies with a shorter than 24 hr period). Finally, flies in which the production of *tim-sc* is abrogated display altered patterns of locomotor activity at 18°C and 25°C, as well as altered expression of the remaining *tim* isoforms, demonstrating the importance of *tim-sc* production. We attribute the partial rescues observed with both TIM-SC and TIM-L to the fact that these proteins were expressed using the GAL4-UAS system, resulting in high levels of TIM. In any case, our results suggest that TIM-SC is functional, although nearly half the size of the canonical TIM protein. Interestingly, the PER binding region of TIM, which is located within the regions from amino acid 715 through amino acid 914 of the canonical protein (*Saez and Young, 1996*), could be compromised in the short isoform, as TIM-SC is only 892 amino acids long. Indeed, we showed that TIM-SC can bind but cannot stabilize PER. It will be interesting to determine the mechanism of action of TIM-SC and to evaluate how this protein interacts with the CLK-PER transcriptional feedback loop. *tim-cold* is strongly regulated by miRNAs at all temperatures. Analysis of the *tim-sc* null mutant provides support for a specific (and even dominant) function of TIM-COLD. We observed that flies that cannot generate *tim-sc* produce more *tim-cold* and, although to a lesser degree, *tim-L* than wild-type flies. These mutant flies also display altered behavior: the flies seem 'hyper-adapted' to cold temperatures with high locomotor activity during the day even at 25°C. It is difficult to rationalize how increased levels of *tim-L* could trigger these cold-like behavioral patterns. We favor the possibility that the behavior observed in these flies at 25°C is due to the increased expression of *tim-cold* mRNA and protein. Previous work showed that TIM-COLD might act differently than TIM-L; indeed it displays weaker binding to CRY in yeast (*Montelli et al., 2015*). Importantly, the increase in TIM-COLD protein in the mutant flies might be significantly larger than the increase observed at the mRNA level. This could happen if, for example, the increase in *tim-L* mRNA indirectly increases TIM-COLD levels by titrating miR-969 or another miRNA that targets both *tim-L* and *tim-cold* mRNAs. Mutations that abrogate *tim-cold* expression or make it constitutive could help to reveal the mechanism behind this regulation.

Although *tim-M* is the most abundant RNA isoform, we did not detect TIM-M protein, suggesting that little or no translation of this RNA occurs. This agrees with recent work by *Shakhmantsir et al. (2018)* who concluded that the major role of *tim-M* was to regulate TIM-L levels. Alternatively, the protein could be quickly degraded. However, this seems unlikely as we detected large amounts of TIM-M upon overexpression, suggesting that the putative protein is very stable. Moreover, our luciferase reporter experiments suggest that little protein is produced from this RNA, probably because of strong miRNA-mediated regulation (as also suggested by the AGO1 immunoprecipitation experiment). miRNAs exert their regulatory role either by promoting destabilization of mRNA or by inhibiting its translation (*Iwakawa and Tomari, 2015*). In the case of *tim-M*, this miRNA-mediated post-transcriptional control most probably occurs through direct translational repression. This would explain the very high mRNA levels of *tim-M* at certain temperatures (sometimes even greater than

those of *tim-L*) and the lack of TIM-M protein expression. It is also possible that this RNA isoform is not exported from the nucleus; nuclear/cytoplasmic fractionation experiments could test this possibility.

Although *tim-cold* and *tim-M* are strongly post-transcriptionally regulated at all temperatures, the RISC also binds strongly to *tim-L* at 18°C. *tim* has been reported to be regulated by miR-276a (*Chen and Rosbash, 2016*); our miRNA-seq experiments suggest that miR-969 might also be important for *tim* regulation. CRISPR experiments targeting the miR-969 site could help to determine this.

Despite the large number of RNAs and miRNAs that are regulated in a temperature-dependent manner, the impact of temperature on alternative splicing seems to be quite restricted. Our analysis of the data revealed that few transcripts other than *tim, per*, and *Hsf* appeared to be regulated by temperature (data not shown). Although splicing efficiency in vitro is strongly temperature dependent, flies might have developed mechanisms (such as compensatory changes in chromatin structure, RNA polymerase II elongation rate, or RNA editing; *Buchumenski et al., 2017*) that make the outcome of alternative splicing largely temperature independent. The specificity of temperature-dependent *tim* splicing strongly suggests that these changes in *tim* splicing are unlikely to be due to the expression or activation of a specific splicing factor. This possibility is supported by the findings of *Foley et al., 2019*, who showed that although *psi* regulates *tim* splicing, this factor on its own cannot explain the temperature sensitivity of these alternative splicing events.

We found that the expression of some miRNAs is affected by temperature, and that levels of several of the temperature-dependent miRNAs correlate with changes in the splicing isoform that they putatively regulate. Hence, the expression of particular miRNAs per se could act as a thermosensor. Our miRNA profiling and AGO1 immunoprecipitation experiments suggest that other mRNAs might also be regulated by miRNAs in a temperature-dependent way. It will be interesting to correlate these datasets that evaluate AGO1-bound miRNA and mRNA expression as a function of temperature in order to understand how the transcriptional and post-transcriptional expression programs are regulated by temperature. Given the central role of RNA structure in the biosynthesis of miRNAs, the production of some miRNAs could be influenced by temperature through effects on secondary structure of the primary transcripts.

The temperature-dependent changes in the abundance of the *tim* isoforms described here are unlikely to be due to regulation by miRNAs, and our data suggest that these changes are mainly due to differences in alternative splicing. First, *tim-sc* does not seem to be regulated by miRNAs at any temperature and still displays large temperature-dependent changes. Second, the AGO1 immunoprecipitation experiments demonstrate a strong post-transcriptional regulation of *tim* at 18°C, when the levels of *tim-cold* are very high. Consequently, the changes in *tim-cold* levels cannot be the result of post-transcriptional control.

Our results demonstrate that the introns in isolation are able to sense and respond to temperature changes. Interestingly, *tim-cold* and *tim-M* respond in opposite ways to changes in temperatures: the intron found in *tim-cold* is strongly retained at 18°C while that in *tim-M* is more frequently skipped at this temperature. These results suggest that RNA structure probably plays a key role in making these splicing events temperature sensitive. Secondary structure prediction revealed no strong temperature-dependent differences. We found that the 5′ splice sites of the introns that are retained in *tim-sc* and *tim-cold* are highly engaged in secondary structures. As with any RNA secondary structure, these structures should be more stable at lower temperature, and this could hinder their recognition by the spliceosome and promote their retention at 18°C. On the other hand, neither the 5′ nor the 3′ splice sites of *tim-M* display strong secondary structures. As canonical splicing is less efficient in fly heads at higher temperatures (S Kadener, unpublished), the retention of this intron will be increased at 29°C despite the weaker 3′ splice site.

On the basis of these and previous results, we propose the model shown in *Figure 9* of the effect of temperature on the expression of each *tim* transcript. This model accounts for the temperature-dependent thresholds for protein expression regulated by miRNA for the different *tim* isoforms. Briefly, at 25°C and 29°C, *tim-L* and *tim-M* isoforms represent the majority of *tim* mRNA, whereas *tim-sc* and *tim-cold* are most abundant at 18°C. *tim-M* is also a target of many miRNAs, some that are not differentially regulated by temperature and others that are more abundant at 29°C than at 18°C, or the opposite. We predict that the threshold imposed by miRNAs is high enough at all tested temperatures that no TIM-M is produced. We postulate that the *tim-M* transcript itself has a regulatory role, as recently suggested by *Shakhmantsir et al. (2018)*. *tim-cold* is under strong

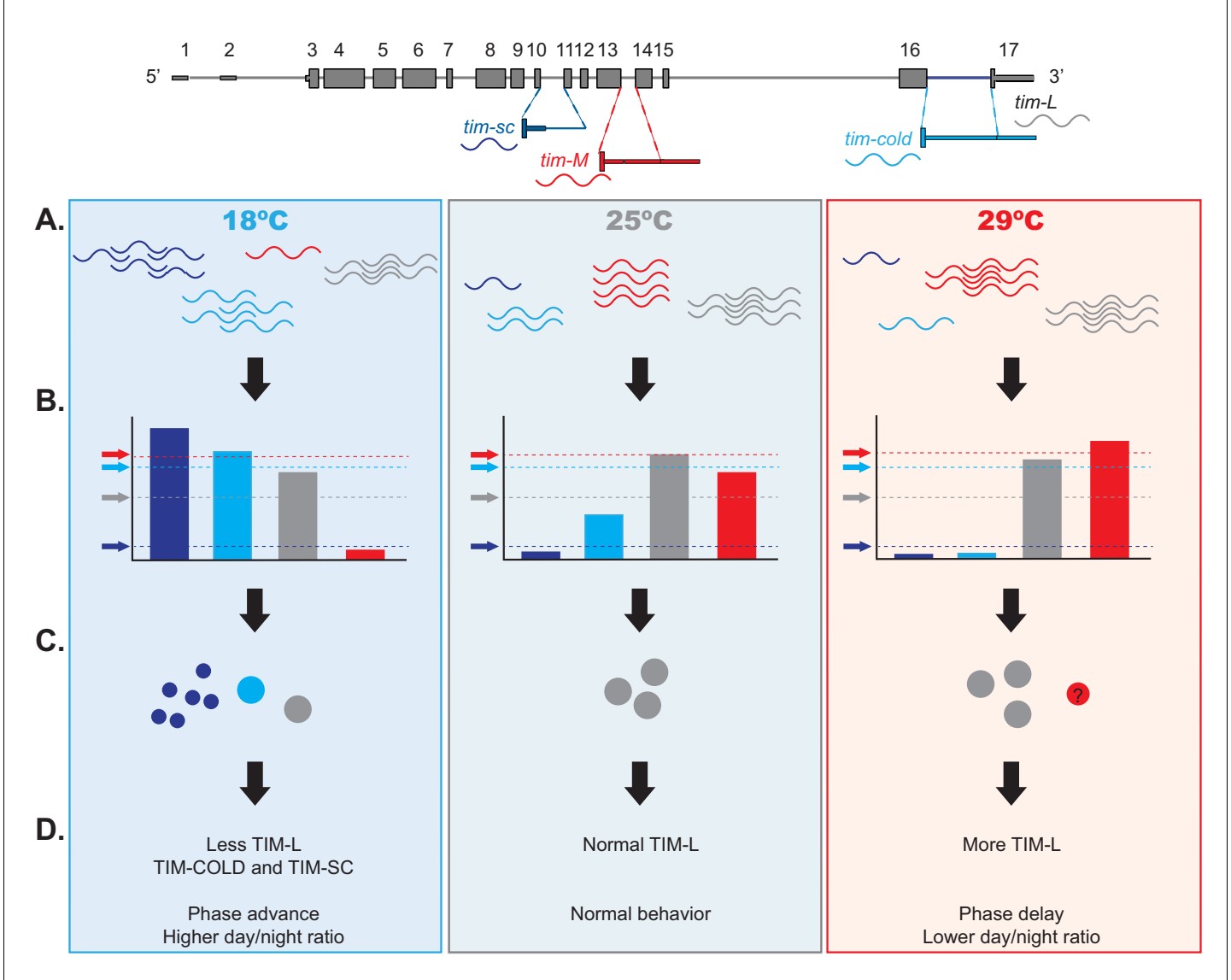

**Figure 9.** Model of the effects of thermosensitive alternative splicing of *timeless* in sensing and mediating temperature adaptation. (**A**) Temperature affects each one of the four *tim* isoforms (*tim-sc, tim-cold, tim-M*, and *tim-L*, represented in dark blue, light blue, red, and gray, respectively) differently. *tim-sc* and *tim-cold* are more abundant at colder temperatures, whereas *tim-M* is expressed at higher temperatures. Abundance of the canonical isoform, *tim-L*, does not change with temperature. (**B**) In addition, miRNAs set the threshold for the amount of mRNA that is translated from each isoform. *tim-sc* has the lowest and *tim-M* the highest threshold at all temperatures. (**C–D**) Differences between the transcript levels and their thresholds result in changes in TIM expression (**C**) and behavioral changes in the phase and day/night ratios (**D**) that are linked to temperature adaptation.

miRNA regulation: at 25°C, when the production of this isoform is low, the miRNA-mediated control is enough to abolish most (if not all) TIM-COLD protein expression. However, at lower temperatures, the strong increase in *tim-cold* overcomes miRNA-mediated repression, and TIM-COLD is produced. In addition, at 18°C, *tim-sc* RNA and protein are produced. We hypothesize that both TIM-COLD and TIM-SC contribute to the phase advance and reduced night activity observed at 18°C. TIM-SC lacks the cytoplasmic retention domain and cannot stabilize PER, so the increased levels of this protein might lead to the phase advance observed at 18°C. At 29°C, *tim-M* levels are higher than at 25°C, but TIM amounts are not altered. This is probably the result of higher translation rates of TIM (*tim-L* association with AGO1 is modestly increased at the lowest temperature). Thus, co- and post-transcriptional regulation of *tim* modulate the amount and type of TIM proteins produced to

facilitate temperature adaptation. We propose that this complex control of alternative splicing and miRNA-mediated control regulates the relationship between TIM and PER proteins. Last, but not least, our data suggest that *tim* alternative splicing might act as a thermometer for the circadian clock.

## Materials and methods

### Fly husbandry

*D. yakuba*, *D. simulans*, and *D. virilis* were obtained from the *Drosophila* Species Stock Center (DSSC) at the University of California, San Diego. CantonS flies were used as wild-type strain for *D. melanogaster*. The $tim^0$ and $per^{01}$ strains have been described previously (*Konopka and Benzer, 1971*; *Sehgal et al., 1994*).

Transgenic lines for the overexpression of different FLAG-tagged *tim* isoforms and luciferase reporters were generated by integrating the plasmids into the attP2 site using the PhiC31 integrase-mediated transgenesis system (Best Gene Drosophila Embryo Injection Services). These transgenic flies were crossed into a *tim-Gal4* driver line (*Kaneko and Hall, 2000*; *Blau and Young, 1999*). These transgenic flies were also crossed with $tim^{01}$ flies to generate $tim^{01}$, *UAS-tim-sc* and $tim^{01}$, and *UAS-tim-L* flies for the rescue experiments. All crosses were performed and raised at 25°C. Newborn adults were either maintained at 25°C or switched to colder (18°C) or warmer (29°C) temperatures, as described in the text.

### Generation of tim-sc polyadenylation and cleavage signal mutants by CRISPR

The *tim-sc* mutant flies were generated following the protocol described by *Ge et al. (2016)* with some modifications. pCFD5 plasmid (Addgene, plasmid #73914) was modified to exclude *Vermilion* and *attB* (pCFD5d). Three gRNAs (one targeting the *w* gene and two targeting *tim*) were generated from PCR templates and cloned into pCFD5d (pCFD5d*w/tim*-1,2) as described (*Port and Bullock, 2016*; *Port et al., 2014*; *Fu et al., 2014*). The donor template for homologous recombination contained a point mutation in an intronic sequence as well as a silent point mutation (one in each site targeted by the gRNAs). This fragment was then cloned into pUC57-*white*[coffee] (Addgene, plasmid #84006) between the *SacI* and *HindIII* sites (pUC57-*timShort*). pCFD5d*w/tim*-1,2, pUC57-*white* [coffee], and pUC57-*timShort* were injected into *vas-Cas9* ($y^1$, M{vas-Cas9ZH-2A) flies (*Ge et al., 2016*) by Rainbow Transgenic Flies, Inc $G_0$ flies were crossed to second chromosome balancers, and individual $G_1Cyo$ flies from non-red-eyed populations were again mated to the 2nd chromosome balancers. Individual $G_1$ flies were genotyped to verify the deletion and the possibility of random integration either in the genome or at CAS9 cutting sites of the plasmid (*Ge et al., 2016*). $G_2Cyo$ male and female flies from positive crosses were crossed to obtain non *cyo* homozygous stocks. Finally, the entire *tim* locus was sequenced in the *tim-sc* mutants and their isogenic controls to verify that there were no additional mutations in the *tim* locus.

### Plasmids for the transfection of S2 cells and/or injection into *Drosophila*
#### Previously described plasmids
pMT-*Clk* (*Weiss et al., 2014*); p-Act-Gal4 (Addgene, plasmid #24344) plasmids were used.

#### C-terminus FLAG-tagged UAS-*tim* overexpression plasmids
cDNA from Canton-S heads of flies reared at 18°C (for *tim-sc* and *tim-cold*), 25°C (*tim-L*), or 29°C (*tim-M*) were used to amplify the ORF of each isoform. *NotI* and *KpnI* restriction sites and a FLAG-tag sequence (at the 3′ of the coding DNA but before the stop codon) were added by PCR. Each isoform was cloned into the pUASTattB vector between the *NotI* and *KpnI* sites.

#### pAcTimL-HA and pAcTimSC-HA plasmids
pAcTimL-HA was a kind gift from Michael Rosbash (Brandeis University). pAcTimSC was cloned by amplifying *tim-sc* ORF from pAcTimL. *NotI* and *KpnI* restriction sites and a HA-tag sequence were added by PCR.

### UAS-luciferase-3′UTR of *tim* isoform plasmids
A PCR product containing the firefly *luciferase* gene was cloned into pUASTattB (pUAST-*luc*-attB). The 3′ UTRs of all four isoforms were amplified from fly head cDNA and cloned into the pUAST-*luc*-attB vector between *NotI* and *XhoI* sites.

### *tim* exon-intron-exon minigenes
Plasmids containing exon 5, intron 5, and exon 6 (*tim-control*); exon 10, intron 10, and exon 11 (*tim-sc*); exon 13, intron 13, and exon 14 (*tim-M*); and exon 16, intron 16, and exon 17 (*tim-cold*) fragments defined by *KpnI* and *XhoI* restriction sites were purchased from Syntezza Bioscience. Plasmids were cut with *KpnI* and *XhoI*. The four fragments were extracted and cloned into pMT-V5 (Invitrogen).

## Cell culture and transfections
*Drosophila* Schneider-2 (S2c1) cells (kindly provided by Michael T. Marr, Brandeis University; FLYB: FBrf0232536; RRID:CVCL_IZ06) were cultured in 10% fetal bovine serum (Invitrogen) insect tissue culture medium (Biological industries) as described (*Weiss et al., 2014*). The experiments were performed at 18°C or 25°C as described in the text. Mycoplasma grows mostly at 30°C or above, hence contamination of S2 cells by mycoplasma is not common.

## Luciferase activity assay
Luciferase measurements from individual flies were performed as previously described (*Glaser and Stanewsky, 2005*) after 16 hr exposure to 10 mM luciferin (syd labs, MB000102-R70170).

## RNA libraries preparation for RNA-seq
### 3′ RNA-seq
Flies were entrained for at least 3 days at 18°C, 25°C, or 29°C and collected at six different time-points (ZT3, ZT7, ZT11, ZT15, ZT19, and ZT23). RNA from the fly heads was extracted using TRIzol reagent (SIGMA). 3′ RNA-seq libraries were prepared as previously described (*Afik et al., 2017*).

### Total RNA-seq
RNA from fly heads collected at two or three different time points (ZT3, ZT15, and ZT21 for 18°C and 29°C, and ZT3 and ZT15 for 25°C) was extracted using Trizol and used to generate polyA+ RNA-seq libraries. The library preparation procedure was modified from that described previously (*Engreitz et al., 2013*) as follows: 0.5 μg of total RNA was polyA+ selected (using Oligo(dT) beads, Invitrogen), fragmented in FastAP buffer (Thermo Scientific) for 3 min at 94°C, then dephosphorylated with FastAP, cleaned (using SPRI beads, Agencourt), and ligated to linker1 (5′-pA XXXXXXXXAGATCGGAAGAGCGTCGTGTAG(ddC)−3′, where XXXXXXX is an internal barcode that is specific for each sample) using T4 RNA ligase I (NEB). Ligated RNA was bound to silane beads (Dynabeads MyOne, Life Technologies), precipitated, and pooled into a single tube. Reverse transcription was then performed with a specific primer (5′-CCTACACGACGCTCTTCC-3′) using an AffinityScript Multiple Temperature cDNA Synthesis Kit (Agilent Technologies). The RNA from the RNA–DNA hybrids was degraded by incubation in 100 mM NaOH at 70°C for 12 min. The pH of the solution was neutralized by the addition of acetic acid to a final concentration of 75 mM. After purification using silane beads, a second ligation was performed, where the 3′ end of cDNA was ligated to linker2 (5′-pAGATCGGAAGAGCACACGTCTG(3ddC)−3′) using T4 RNA ligase I. The sequences of linker1 and linker2 are partially complementary to the standard Illumina read1 and read2/barcode adapters, respectively. After purification using silane beads, PCR enrichment was performed using primers 1 and 2: (5′-AATGATACGGCGACCACCGAGATCTACACTCTTTCCC TACACGACGCTCTTCCGATCT-3′, 5′-CAAGCAGAAGACGGCATACGAGATXXXXXXXXGTGAC TGGAGTTCAGACGTGTGCTCTTCCGATCT-3′, respectively, where XXXXXXX is barcode sequence) and Phusion HF MasterMix (NEB). Twelve cycles of enrichment were performed. Libraries were purified with 0.7X volume of SPRI beads and characterized using Tapestation. RNA was sequenced as paired-end samples in a NextSeq 500 sequencer (Illumina).

## AGO1 immunoprecipitation followed by small RNA sequencing

The samples were collected at 18°C, 25°C, or 29°C at four different time points (ZT3, ZT9, ZT15, and ZT21). AGO1 immunoprecipitation was performed as previously described (*Kadener et al., 2009*; *Lerner et al., 2015*). The small RNA libraries were constructed using NEBNext Small RNA Library Prep Set for Illumina.

## AGO1 immunoprecipitation followed by oligonucleotide microarrays

The samples were collected at 18°C, 25°C, or 29°C. AGO1 immunoprecipitation was performed as previously described (*Kadener et al., 2009*; *Lerner et al., 2015*). Input and AGO1 IP Total RNA was extracted from fly heads using TRI reagent (Sigma) according to manufacturer's protocol. cDNA synthesis was carried out as described in the Expression Analysis Technical Manual (Affymetrix). The cRNA reactions were carried out using the IVT kit (Affymetrix). Affymetrix high-density arrays for *D. melanogaster* version 2.0 were probed, hybridized, stained, and washed according to the manufacturer's protocol.

## Microarray analysis

The R Bioconductor affy package (http://www.bioconductor.org) was used to normalize and calculate summary values from Affymetrix CEL files using gcRMA (Bioconductor). Gene ontology enrichment analysis was performed as described in *Mezan et al. (2013)*. To determine the enrichment ranking, we ranked the residuals from a linear model. AGO1 IP enrichment was determined by dividing the normalized reads in the IP and the input.

## RNA-seq analysis

The total RNA sequencing reads were aligned to the *D. melanogaster* genome version dm3. Alternative splicing proportion was calculated manually by searching for the exon–exon junction.

For the small RNA sequencing, the reads were processed using miRExpress pipeline (*Wang et al., 2009*) using miRBase 21 version. The different circadian time points were considered to be independent replicates (n = 4), and the differential gene expression analysis was done using a negative binomial model employing DeSeq2 package on R. A miRNA with p-adjusted value less than 0.05 and an absolute $\log_2$(fold change) more than one was considered differentially expressed.

For the 3′ RNA-seq data, the circadian analysis was performed using the package MetaCycle (*Wu et al., 2016*). For each temperature and circadian time point, two replicates were analyzed. To normalize over different library preparations, after normalizing by library size, the counts were divided by the maximum in each replicate. Genes with more than two zero counts in any time point were discarded. The amplitude for each replicate was then calculated as the maximum divided by the minimum for each gene. The JTK algorithm was used for the circadian analysis (*Hughes et al., 2010*). A gene was considered to be cycling if the JTK p-value was less than 0.05 and the amplitude was more than 1.5.

## Assessment of splice site strength

The splice strength was determined using a publicly available software (*Low et al., 2008*; *Reese et al., 1997*). The sequences for *tim* were downloaded from the UCSC genome browser.

## RNA secondary structure at different temperatures

We used the Mfold web server version 2.3 to predict the secondary structures of the 3′ UTRs of each *tim* isoform (*Zuker, 2003*). For each isoform, we selected the relevant intron as well as the flanking exons. Owing to limitations in the length of the sequence that can be uploaded, for *tim-cold* we used 350 bases from each flanking exon. RNA sequence was downloaded from UCSC fly genome version dm6. The maximum interior/bulge loop size and maximum asymmetry of an interior/bulge loop was set to 30.

## TargetScan analysis of putative miRNA binding to each tim 3′UTR

As not all of the 3′ UTRs for the different *tim* isoforms are annotated, TargetScanFly version 6.1 was run locally (which allows manual entry of the sequences of interest). UTR sequences were

downloaded from UCSC dm6 27way conservation. We identified several miRNAs that have the potential to bind to each of the isoforms. We then determined the abundance of these miRNAs from the AGO1 immunoprecipitates followed by small RNA sequencing in order to determine which of those putative miRNAs were expressed in the fly heads. In addition, we defined miRNAs that change in a temperature-dependent manner as those miRNAs that had more than a two-fold, statistically significant (p<0.05), change between 18°C and 29°C.

## Chromatin-bound RNA

Chromatin-bound RNA was isolated as described in *Lerner et al. (2015)*.

## Real-time qRT-PCR analysis

Total RNA was extracted from adult fly heads (or brains) at the mentioned time points using TRI Reagent (Sigma) and treated with DNase I (NEB) following the manufacturer's protocol. cDNA was synthesized from this RNA (using iScript and oligodT primers, Bio-Rad) and diluted 1:60 prior to performing the quantitative real-time PCR using SYBR green (Bio-Rad) in a C1000 Thermal Cycler Bio-Rad. Primers used for amplifying each isoform were: *tim*-sc (5′-AACACAACCAGGAGCATAC-3′ and 5′-ATGGTCCACAAATGTTAAAA-3′), *tim*-M (5′-GGAGACAATGTACGGACTC-3′ and 5′-ATTTCACA-CAGAGAGAGAGC-3′), *tim*-cold (5′-GCATCTGTGTACGAAAAGGA-3′ and 5′-ATGTAACCTATG TGCGACTC-3′), *tim*-L and *tim*-M (5′-CTCCATGAAGTCCTCGTTC-3′ and 5′-TGTCGCTGTTTAATTCC TTC-3′), and junction between exons 5–6 which are present in all isoforms (5′-AAAAGCAGCCTCA TCAACAT-3′ and 5′-AGATAGCTGTAACCCTTGAG-3′).

The PCR mixture was subjected to 95°C for 3 min, followed by 40 cycles of 95°C for 10 s, 55°C for 10 s, and 72°C for 30 s, followed by a melting curve analysis. Fluorescence intensities were plotted against the number of cycles by using an algorithm provided by the manufacturer (CFX Maestro Software, Bio-Rad). The results were normalized against *rp49* (5′-TACAGGCCCAAGATCGTGAA-3′ and 5′-CCATTTGTGCGACAGCTTAG-3′) and *tub* (5′-TGCTCACGAAAAGCTCTCCT-3′ and 5′-CACACACGCACTATCAGCAA-3′) levels.

## PER and HA immunoprecipitations

S2 cells transfected with pAcPerV5, pAcTim-L-HA, and/or pAcTim-SC-HA were harvested 48 hr post-transfection. Cells were washed with 1xPBS and resuspended in 100 μl of lysis buffer (1% NP40, 20 mM Tris, pH 7.4, 150 mM NaCl, 10% glycerol, proteinase inhibitor [Roche]). The lysates were incubated for 15 min on ice and centrifuged at 14,000 rpm for 15 min at 4°C. A aliquot of 10 μl of the supernatant was used as input. The rest of the supernatant (~90 μL) was incubated with 10 μg of α-PER or α-HA for 2 hr at 4°C with rotation. Afterwards, 45 μL of Protein G Magnetic beads (S1430S, NEB) were added for 1 hr at 4°C with rotation. Using a magnetic stand, the beads were washed three times with lysis buffer, before analysis by western blot.

## Western blotting

Protein was extracted from fly heads and western blot was performed as described (*Weiss et al., 2014*). The antibodies used were as follows: rat anti-TIM (1:30,000, a kind gift from Michael Rosbash), mouse anti-FLAG-M2 (1:20,000; Sigma F3165), rabbit anti-PER (1:1,000, a kind gift from Paul Hardin, Texas A and M), rabbit anti-GFP (1:1,000, Genesee), mouse anti-HA (1:1,000, Roche), and mouse anti-Tub (1:20,000, DM1A, Sigma). Quantifications were done utilizing Image J software.

## Assessment of the locomotor behavior

Male flies were placed into a glass tube containing 2% agarose and 5% sucrose food, and their activity was monitored using Trikinetics *Drosophila* Activity Monitors. Flies were maintained for 4 days in 12:12 LD and 5 days in DD. The experiments were performed at 18°C, 25°C, or 29°C as indicated. Analyses were performed with a signal processing toolbox (*Levine et al., 2002*).

All the activity assessments were done in LD. For the calculation of the beginning of the evening and morning activity component, the activity of flies was analyzed individually at 30 min intervals. The criteria were: (1) a trend of increasing activity, (2) no more than 1/3 of the bins can be outliers, and (3) a reduction of less than 10% of the activity was not counted as an

outlier. Only flies in which an evening or morning activity component was clearly distinguishable were included in this analysis.

## Acknowledgements

We thank Avigayel Rabin and Reut Ashwall-Fluss for help with the alignment of the RNA-seq data. This work was funded by the National Institute of Health under award number R01GM125859 to SK. All of the RNA-seq data have been submitted to GEO (entries GSE124134, 124135, 124141, 123142, 124200, and 124201).

## Additional information

### Funding

| Funder | Grant reference number | Author |
|---|---|---|
| National Institutes of Health | R01GM125859 | Sebastian Kadener |

The funders had no role in study design, data collection and interpretation, or the decision to submit the work for publication.

### Author contributions

Ane Martin Anduaga, Data curation, Investigation, Visualization, Methodology, Writing—original draft, Project administration, Writing—review and editing; Naveh Evantal, Conceptualization, Investigation, Visualization, Methodology, Writing—original draft; Ines Lucia Patop, Conceptualization, Data curation, Formal analysis, Visualization; Osnat Bartok, Investigation, Methodology; Ron Weiss, Investigation, Visualization; Sebastian Kadener, Supervision, Funding acquisition, Investigation, Writing—original draft, Project administration, Writing—review and editing

### Author ORCIDs

Ane Martin Anduaga (iD) https://orcid.org/0000-0003-2447-2195
Sebastian Kadener (iD) https://orcid.org/0000-0003-0080-5987

### Decision letter and Author response

Decision letter https://doi.org/10.7554/eLife.44642.sa1
Author response https://doi.org/10.7554/eLife.44642.sa2

## Additional files

### Supplementary files

• Supplementary file 1. Circadian oscillation analysis.The circadian analysis was performed using the MetaCycle algorithm. In the table, the p-value, phase, and amplitude are reported for each gene along with its normalized expression at each circadian time point.

• Supplementary file 2. Gene ontology analysis for the genes with circadian oscillation at each temperature.Genes are ordered by ranking in a Fisher exact test. Three different ontologies were analyzed: biological process, molecular function, and cellular component. The top 100 genes are reported.

• Supplementary file 3. miRNA binding sites predicted using TargetScan for each *tim* RNA isoform 3′ UTR.Putative miRNA binding sites are reported as chromosome coordinates. Conservation in different insect species is also reported.

• Supplementary file 4. Differential miRNA expression at each temperature.Log$_2$(fold change) and adjusted p-value from DeSeq2 are reported for each miRNA.

• Transparent reporting form

## Data availability

Sequencing data have been deposited in GEO under accession codes GSE124134, 124135, 124141, 123142, 124200 and 124201.

The following datasets were generated:

| Author(s) | Year | Dataset title | Dataset URL | Database and Identifier |
|---|---|---|---|---|
| Martin Anduaga A, Evantal N, Patop IL, Bartok O, Weiss R, Kadener S | 2019 | Small RNA seq from Ago1-IP of fly heads at different temperatures and circadian timepoints | https://www.ncbi.nlm.nih.gov/geo/query/acc.cgi?acc=GSE124134 | NCBI Gene Expression Omnibus, GSE124134 |
| Martin Anduaga A, Evantal N, Patop IL, Bartok O, Weiss R, Kadener S | 2019 | total mRNA seq from fly heads at different temperatures and circadian timepoints | https://www.ncbi.nlm.nih.gov/geo/query/acc.cgi?acc=GSE124135 | NCBI Gene Expression Omnibus, GSE124135 |
| Martin Anduaga A, Evantal N, Patop IL, Bartok O, Weiss R, Kadener S | 2019 | 3' end RNA seq from fly heads at different temperatures and circadian timepoints | https://www.ncbi.nlm.nih.gov/geo/query/acc.cgi?acc=GSE124141 | NCBI Gene Expression Omnibus, GSE124141 |
| Martin Anduaga A, Evantal N, Patop IL, Bartok O, Weiss R, Kadener S | 2019 | Total RNA from different species fly heads | https://www.ncbi.nlm.nih.gov/geo/query/acc.cgi?acc=GSE124142 | NCBI Gene Expression Omnibus, GSE123142 |
| Martin Anduaga A, Evantal N, Patop IL, Bartok O, Weiss R, Kadener S | 2019 | 3' end RNA seq from fly heads at different temperatures and circadian timepoints | https://www.ncbi.nlm.nih.gov/geo/query/acc.cgi?acc=GSE124200 | NCBI Gene Expression Omnibus, GSE124200 |
| Martin Anduaga A, Evantal N, Patop IL, Bartok O, Weiss R, Kadener S | 2019 | Expression data from Drosophila heads at different temperatues, after AGO1 immnuprecipitation or input [microarray] | https://www.ncbi.nlm.nih.gov/geo/query/acc.cgi?acc=GSE124201 | NCBI Gene Expression Omnibus, GSE124201 |

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
