## [Decision Letter]

**Acceptance summary:**

In poikilotherms (animals whose internal temperature varies considerably) the circadian clock must tick at the appropriate 24 h period in the different thermal environments that these organisms inhabit. This temperature adaptation has yet to be understood in terms of mechanisms that convey environmental signals to the conserved core oscillator composed of PER, TIMELESS, CLOCK and CYCLE genes in *Drosophila melanogaster*, whose individual activities and stoichometry are critical for the period of oscillation, previous work (e.g. Majercak et al., 1999) has implicated temperature dependent splicing of Per as playing a role in seasonal temperature adaptation and more recent observations have shown the involvement of alternative splicing of TIMELESS (e.g. Boothroyd et al., 2007; Shakhmantsir, 2018) in the control of circadian rhythms. In this work, Anduaga et al. examine the relationship between TIM splice isoforms and seasonal temperature-dependent changes in daily locomotor activity. In particular, Anduaga et al. show and characterize the expression of two 19 °C cold-specific splice isoforms: *tim-cold* and *tim-short&cold* as well as a 29 °C warm temperature-selective isoform, tim-medium, with potentially different functions and activities. Key aspects of this alternative splicing pattern are conserved across non-equatorial *Drosophila* species. Further analyses address how these splice isoforms may affect TIM activity and argue that miRNAs could confer isoform-selective, post-transcriptional control of TIM expression. Thus, the work provides interesting new insight to how circadian rhythms remain constant across temperatures.

**Decision letter after peer review:**

Thank you for submitting your article "Thermosensitive alternative splicing senses and mediates temperature adaptation in *Drosophila*" for consideration by *eLife*. Your article has been reviewed by two peer reviewers, and the evaluation has been overseen by Mani Ramaswami as Reviewing Editor and Ronald Calabrese as the Senior Editor. The reviewers have opted to remain anonymous.

The reviewers have discussed the reviews with one another and the Reviewing Editor has drafted this decision to help you prepare a revised submission.

Summary:

This interesting manuscript represents a substantial amount of work analysing mechanisms that allow temperature adaptation of circadian behavior in *Drosophila*. The authors examine the relationship between TIM splice isoforms and seasonal temperature-dependent changes in daily locomotor activity, and so demonstrate that cold temperature (18 °C) downregulates the canonical *tim* transcript (*tim-L*) mRNA isoform. More interestingly, the authors report a concomitant induction of two cold-specific splicing isoforms [*tim-short and cold (sc*) and *tim-cold*]. Another isoform *tim-M* is present at high levels at all temperatures and is likely non-functional. TIM-sc is generated by use of an alternative polyA site; TIM-cold (previously reported) generates a shorter protein as it is an intron retained transcript (with premature stop) with a longer 3'UTR. Previous to this study, a number of papers had been published on *per* 3' splicing in the context of thermal adaptation of circadian locomotor rhythms by Edery's group. Some work has already been performed on *tim-cold* and *tim-M (tim-tiny*) by other groups. The more original results here relate to *tim-sc*, as well as its regulation by miRNAs.

The TIM -sc isoform is conserved and appears only at 18 °C in both *D. simulans* and *D. melanogaster*, strongly suggesting a role in adaptation to cold temperatures. At the same time, TIM protein levels are reduced. Taking leads from other work (Chen and Rosbash, 2016) the authors examine post-transcriptional regulation by miRNAs. Interestingly, AGO1 associated spliced isoforms clearly showed low-temperature dependent association of Tim-isoforms. Using cell-culture based luciferase reporter assays the 3'UTR of Tim-sc alone was shown to be under strong miRNA dependent regulation. Further, the authors carried out functional studies of locomotor behavior with flies overexpressing *tim-sc* (using *tim-Gal4* driver) and make interesting observation of advanced phase in evening locomotor behavior. The authors examined Crisper edited lines that abrogate processing of the *tim-sc* form. These mutants had a significant delay, at 18 °C, in the onset of evening activity, a phenotype also seen mildly at 25 °C. These data clearly show the functional significance of cold-based induction of Tim-sc isoform. Finally, they investigate the possibility that *tim* splicing per se functions as a thermometer for seasonal dependent adaptation and infer that the intronic sequences of *tim* serve as a thermosensor to modulate the temperature-specific splicing changes.

Essential revisions:

The major scientific issues to be addressed pertain to: (a) the need to more clearly establish the function of the *tim-sc* isoform. And (b) deeper consideration of the mechanism by which the *tim-sc* splicing event occurs. In addition, the manuscript could be tightened up and needs to be edited to improve readability and also grammar on occasion.

1) To test whether the *tim* isoforms are functional they should be put into a *tim^0^* background to test whether they rescue rhythmicity at different temperatures. This is quite fundamental to the story. Sehgal's group have shown that *tim-M* partially rescues the mutant and *tim-cold* also rescues. It would be interesting to see what *tim-sc* does in a *per^0^* background in LD and DD. It is not clear why this has not been done.

2) The model that temperature specificity of *tim* splicing arises from an intron serving as thermosensor is a particularly interesting aspect of the paper. It would be informative to present and discuss the predicted RNA secondary structures of exonic and intronic sequences involved in various *tim* alternative splicing events. What are the predicted secondary structure changes that occur on transitions from 18, 25 and 29 °C. Further, whether and which features of secondary structure change with polyadenylation and cleavage signal mutations (used for CRISPR based abrogation of *tim-sc*) in intron 10. An attempt should also be made (e.g. via mini-gene exon-intron constructs in S2 cells) to more definitively test if *tim* exon-introns themselves confer temperature dependent splicing pattern.

3) Considering the alternatively spliced isoform (*tim-sc*) codes for a functional protein, (subsection “Elimination of *tim-sc* results in changes in *tim* processing and locomotor activity”, last paragraph) the implications for its protein interactions are not clearly elucidated. Are the domains important for interaction with PER affected due to the alternative splicing? Is binding and stabilization of PER, and its migration to the nucleus normal?

Less critical issues:

4) General statistical point: the authors prefer to use multiple t-testing with the Holm-Sidak correction. Several analyses might be better served with ANOVA (one or two-way) with post hoc tests – far more elegant and informative for main effects and interactions, particularly in a two-way analysis, e.g. for the data in Figure 6D and E and possibly even 3-way for 6F.

5) Subsection “Temperature remodels the circadian transcriptome”, first paragraph. The authors used JTK cycle for analysis. What was the cut-off for significance and was a correction for multiple testing employed? I can't find this in the Materials and methods.

6) Subsection “Temperature remodels the circadian transcriptome”, first paragraph. Figure 1A – not obvious that cycling phase is earlier in the cold from these panels.

7) Subsection “Temperature remodels the circadian transcriptome”, first paragraph. Figure 1C earlier phase of clock genes – some of those clock genes do not show significant cycling – I had a quick look at the raw data in Supplementary file 1, so I am not sure what this figure really represents….It's only OK if you just include, *per, tim* and *vri* which cycle robustly.

8) Subsection “Cold temperature decreases TIM-L levels by two independent mechanisms”, first paragraph. Shakhmantsir et al., 2018, et al. may have seen a band on a western corresponding to *tim-M* (see their Figure 5B) so it does encode a TIM product.

9) Figure 4C and subsection “Cold temperature decreases TIM-L levels by two independent mechanisms”, third paragraph. When the authors state that LUC reporter levels were high – compared to what? This suggests not much miRNA involvement for *tim-sc*. However as the endogenous *tim-sc* levels in the flies head are low at 25 °C (Figure 2C), does this mean that these low levels are little to do with miRNAs destabilising the transcript?

10) Subsection “Overexpression of the different TIM isoforms alters circadian behavior differently”, first paragraph. Is there also a corresponding advance in the M component in *tim-sc* –.…certainly looks like it – maybe also in *tim-cold*?

11) Subsection “Elimination of *tim-sc* results in changes in *tim* processing and locomotor activity”, first paragraph. Some of the descriptions of the actograms are a bit subjective. I would have described the most obvious difference is the very strong siesta in the mutant at 29 °C. This is not mentioned. Also it appears that there is a reduction in M anticipation of lights on. One striking feature of Figure 6C is the lack of variation in the 40A profiles which makes it quite difficult to compare 40A with the control as the latter profiles seem so sloppy.

12) The authors conclude that *tim-M* and *tim-cold* are strongly post transcriptionally regulated through miRNAs at all temperatures. While this matches with expression levels for *tim-cold* in Figure 2C, it is unclear with regard to *tim-M* which is relatively highly expressed at all temperatures often higher than the canonical isoform (18<25<29). This needs to be explained.

[Editors' note: further revisions were requested prior to acceptance, as described below.]

Thank you for submitting your article "Thermosensitive alternative splicing senses and mediates temperature adaptation in *Drosophila*" for consideration by *eLife*. Your article has been reviewed by two peer reviewers, and the evaluation has been overseen by Mani Ramaswami as Reviewing Editor and Ronald Calabrese as the Senior Editor. The reviewers have opted to remain anonymous.

The reviewers have discussed the reviews with one another and the Reviewing Editor has drafted this decision to help you prepare a revised submission.

The reviewers note that work linking generation of a 18C-specific, timeless splice isoform Tim-Sc to temperature-adaption of the circadian clock has been improved and tightened by several additional experiments. New analysis of Tim-Sc Per interactions and as well as phenotypic effects of *tim-sc* expression in *per^0^* flies are consistent with that Tim-sc triggered, PER independent (but yet unspecified) mechanisms in temperature-adaptation of circadian clock.

However, the manuscript requires extensive text revisions and some simple experimental additions before it can be considered acceptable for publication. The text is dense, difficult to read and appears to have been hastily assembled and needs a fair bit of work to make it more reader-friendly. It also contains several errors including an incorrect description of *tim-sc* which is fundamental to this manuscript and missing references. Further, it includes statements that are not supported by the data. Several very general claims about the *tim* splicing representing general cellular thermometers go way beyond the results – when a simple experiments exist to determine whether this may be true or not. Several, lines of changes are required. First, the manuscript needs to be carefully edited for clarity, grammar and accuracy to be easily accessible and comprehensive to the reader. And second, particularly given that the conclusions, while interesting are not decisive with respect to the mechanism by which Tim-Sc works, some key behavioral experiments (comments 12, 13) should also be conducted at colder temperatures (18 °C) to better test implications that the RNA modelling suggests for stability changes with temperature.

Major comments:

1) The opening sentence makes no sense – ' degradation of genes'? Maybe 'clock gene products'?

2) 'little is known…’ appears to be a bit harsh? Others who work in the field such as Ralf Stanewsky and Isaac Edery may disagree with this and perhaps their insights should be acknowledged.

3) Introduction, third paragraph. TIM is not the key factor, it's a key factor.

4) Introduction, third paragraph. *tim* is the key factor for light signalling' not 'TIM is the key pathway for conveying external information to the circadian system at the molecular level.' The authors are overplaying the role of TIM in all environmental inputs, light, temperature, social, etc. This is incorrect…

5) Introduction, fourth paragraph. Majercak et al., 2004, should also be cited along with Collins et al., 2004.

6) Introduction, fourth paragraph. The Edery group has done a lot of additional work on *per* splicing which has clarified somewhat how *per* splicing might impinge on thermal adaptation. Should be elaborated a little here.

7) Most figures – the authors need to state what the error bars represent -SEMS?

8) Subsection “Temperature modulates *tim* alternative splicing”, first paragraph – there is a 3 fold increase at low temperature of *per* spliced, so *tim* is not the only clock gene with a significant thermal splicing pattern. Also Montelli et al., 2015, showed that at 10 °C *per* spliced reflects about 50% of total transcript and *tim-cold* about 90% of *tim* transcript at this temperature even though overall *tim* levels are low. It is surprising that this highly relevant study is not even mentioned or cited. We note that it is mentioned in the third paragraph of the Discussion in a different context but does not appear in the references.

9) 'In contrast to the other isoforms which are generated by intron retention, *tim-sc* mRNA is generated by the usage of an alternative cleavage and polyadenylation site located within the intron 10 of *tim* which can only be used before this intron is spliced out.'

10) Apart from being ungrammatical, this sentence makes no biological sense. The authors suggest that the intron is spliced out. If so how can the STOP, cleavage and polyA sites be used? Surely this intron or part of it must also be retained?

11) 'As mentioned above, this isoform appears only at 18 °C, strongly suggesting that this isoform is responsible for the adaptation to cold temperatures'. This is an intuitive leap based on a correlation – no causation can be implied at this stage. If this was the case would not one expect higher levels of *tim-sc* and *tim-cold* in *D. virilis* at colder temperatures? So why is it 'surprising' that *D. virilis* does show higher levels of *tim-cold* at lower temperatures Doesn't the presence of *tim-cold* in all species at higher levels at colder temperatures argue that this is the conserved, important transcript for cold adaptation?

12) Figure 5 – Why was the behavioral analysis nor also performed at 18 °C given that all this is done in a WT background? This needs to be done and included and the findings discussed.

13) Figure 6A to C. Again, might it not be useful to examine rescue of *tim^01^* at lower temperature? Perhaps the different UTRs are more or less stable at different temperatures? This appears to be worth a shot perhaps given the last section on RNA structure?

14) 'These results demonstrate that despite the smaller size, *tim-sc* encodes a fully functional TIM protein'. Why 'fully' with only 42% rescue?

15) 'Control flies presented two different responses when transferred to 18 °C: a delay of the evening component as well as a decreased night activity in comparison to the flies maintained at 25 °C '(Figure 6C, 6D and 6E).' Do the authors mean Figure 7C, D, E.

16) 'In addition, the activity patterns of 40A flies are strikingly similar at the different temperatures.' Not really, and no more similar than controls. If one examines panels 7D and 7E the two phenotypes change more in 40A than in controls.

17) 'Last but not least, our data suggest that *tim* alternative splicing might act as a thermometer for the cell.' This is a very general statement, not even limited to the circadian clock. If this is the case then other temperature dependent phenotypes should also be affected by the *tim*-splicing mutants. Why don't the authors check one of these out, for example the temperature at which temperature-sensitive mutants become paralysed e.g. shits, parats etc? Simple enough to do but unlikely to support the authors general claim.

---

## [Author Response]

Essential revisions:The major scientific issues to be addressed pertain to: (a) the need to more clearly establish the function of the tim-sc isoform. And (b) deeper consideration of the mechanism by which the tim-sc splicing event occurs. In addition, the manuscript could be tightened up and needs to be edited to improve readability and also grammar on occasion.1) To test whether the tim isoforms are functional they should be put into a tim^0^ background to test whether they rescue rhythmicity at different temperatures. This is quite fundamental to the story. Sehgal's group have shown that tim-M partially rescues the mutant and tim-cold also rescues. It would be interesting to see what tim-sc does in a per^0^ background in LD and DD. It is not clear why this has not been done.

We thank the reviewers for this suggestion. To address this issue, we determined whether TIM-SC can rescue the behavioral rhythms of *tim^01^* flies. We found that expression of UAS-*tim-sc* using the *Tim-*Gal4 driver rescues the *tim^01^* phenotype in LD (Figures 6A, C). Additionally, we observe a partial rescue (~40% of the flies) in DD. Interestingly, we found that the flies in which locomotor rhythms are rescued, the period is shorter than 24hs (23 h, Figure 6B-C). These results demonstrate that despite the significantly shorter size, TIM-SC is functional.

In addition, and as proposed by the reviewers, we overexpressed *tim-sc* and *tim-L* in *per^01^* flies. We observed an evening component in a small percentage of the flies (18%) suggesting that TIM-SC can act in a PER-independent fashion. In any case the flies remain arrhythmic in DD (Figure 6—figure supplement 2).

2) The model that temperature specificity of tim splicing arises from an intron serving as thermosensor is a particularly interesting aspect of the paper. It would be informative to present and discuss the predicted RNA secondary structures of exonic and intronic sequences involved in various tim alternative splicing events. What are the predicted secondary structure changes that occur on transitions from 18, 25 and 29 °C. Further, whether and which features of secondary structure change with polyadenylation and cleavage signal mutations (used for CRISPR based abrogation of tim-sc) in intron 10. An attempt should also be made (e.g. via mini-gene exon-intron constructs in S2 cells) to more definitively test if tim exon-introns themselves confer temperature dependent splicing pattern.

As suggested, we have now utilized a software to predict the secondary structure of the temperature sensitive introns and their flanking exons. Interestingly, we found that the 5’ splice site in the introns whose splicing prevents the generation of *tim-sc* or *tim-cold* are strongly engaged in secondary structures. We postulate that at lower temperatures these structures are more stable (as any RNA secondary structure) and hence less accessible for recognition by the spliceosome. This could explain why these introns are not recognized as such at 18 °C and remain in the mRNA. Interestingly, the cleavage and polyadenylation signal required for the generation of the *tim-sc* isoform is not engaged in secondary structure. On the other hand, the predicted RNA structures of the intron contained in *tim-M* suggest that neither the 5’ or 3’ splice site are engaged in secondary structure. As splicing efficiency is generally increased at lower temperatures, it is not surprising that this intron will be more skipped at 29 °C, leading to larger amounts of *tim-M* RNA. Last, we also showed that the predicted RNA structure of the *tim-sc* intron and flanking exons does not change by the deletion of the cleavage and polyadenylation site sequence in the intron. We believe that these assessments together with the minigene experiments make a strong case for the possibility that the introns themselves act as thermometers. We thank the reviewer for suggesting solving this issue and have included the predicted structures as well as this explanation in the new version of the manuscript (Figure 8—figure supplement 2).

3) Considering the alternatively spliced isoform (tim-sc) codes for a functional protein, (subsection “Elimination of tim-sc results in changes in tim processing and locomotor activity”, last paragraph) the implications for its protein interactions are not clearly elucidated. Are the domains important for interaction with PER affected due to the alternative splicing? Is binding and stabilization of PER, and its migration to the nucleus normal?

We thank the reviewers for bringing this up. Two regions in TIM are known to be crucial for interacting with PER: one that includes the nuclear localization signal and the other one between amino acid 715 and amino acid 914 of the canonical TIM protein (Saez and Young, 1996). As *tim-sc* encodes a truncated version of the TIM protein (892 aalong), it is possible that the ability of TIM-SC to bind or interact with PER is compromised. We therefore tested whether TIM-SC can bind and stabilize PER. Indeed, in experiments performed in S2 cells, we showed that while TIM-L strongly stabilizes PER (as previously reported), TIM-SC expression does not result in PER stabilization. In addition, we performed co-immunoprecipitation experiments and found that both TIM-L and TIM-SC can bind to PER. Moreover, TIM-SC binding seems to be strong as it can persist in presence of TIM-L. These results show that TIM-SC acts by a different mechanism which involves PER binding but not PER stabilization. We have added these experiments to the new version of the manuscript (Figure 6 and Figure 6—figure supplement 1). We thank the reviewers for bringing this up as it has really improved the manuscript.

Less critical issues:4) General statistical point: the authors prefer to use multiple t-testing with the Holm-Sidak correction. Several analyses might be better served with ANOVA (one or two-way) with post hoc tests – far more elegant and informative for main effects and interactions, particularly in a two-way analysis, e.g. for the data in Figure 6D and E and possibly even 3-way for 6F.

We thank the reviewers for this suggestion. Following their advice, we have reanalyzed the data in Figure 3A, 4B, 6D, 6E, 6F, 7B and 7D using Two-way ANOVA. We incorporated these results in the new version of the manuscript. Using these tests, we obtained similar results to the ones mentioned in the original manuscript.

5) Subsection “Temperature remodels the circadian transcriptome”, first paragraph. The authors used JTK cycle for analysis. What was the cut-off for significance and was a correction for multiple testing employed? I can't find this in the Materials and methods.

We thank the reviewer for pointing this out. We indeed stated this on the methods originally but maybe the reviewer missed it. Briefly, a gene was considered as cycling if the JTK p-value was less than 0.05 and the amplitude was more than 1.5. In addition, we have now cited Hughes et al., 2010, in the Materials and methods, where it is stated that the JTK test performs a Bonferroni-adjustment for correcting for multiple testing.

6) Subsection “Temperature remodels the circadian transcriptome”, first paragraph. Figure 1A – not obvious that cycling phase is earlier in the cold from these panels.

The reviewers are correct, cycling phase differences are not evident in these graphs. The point we are trying to convey is that there are rhythmic transcripts at all temperatures. However, as you can see in the Venn diagram in Figure 1B, there are not many transcripts that are cycling at all temperatures, which makes this advance on the cycling impossible to see unless you focus in particular genes as it is done in Figure 1C. We have now changed this in the manuscript.

7) Subsection “Temperature remodels the circadian transcriptome”, first paragraph. Figure 1C earlier phase of clock genes – some of those clock genes do not show significant cycling – I had a quick look at the raw data in Supplementary file 1, so I am not sure what this figure really represents….It's only OK if you just include, per, tim and vri which cycle robustly.

We understand the point raised by reviewers. We originally included *Clk* and *cry* in this analysis but they are not as strong cyclers as *per, tim* or *vri* which could cause confusion. We have now limited the analysis to the strong cyclers (as suggested) and observe a clear advance in their peak of expression at 18 °C compared to 25 °C (Figure 1C).

8) Subsection “Cold temperature decreases TIM-L levels by two independent mechanisms”, first paragraph. Shakhmantsir et al., 2018, may have seen a band on a western corresponding to tim-M (see their Figure 5B) so it does encode a TIM product.

We thank the reviewers for bringing this up. Indeed, Shakhmantsir et al., 2018 describe a band that corresponds to TIM-M when overexpressing TIM^TINY^ (reported as TIM-M throughout our manuscript), but not in the control, suggesting that there not endogenous TIM-M protein. In any case, we favor the hypothesis in which this transcript is under strong post-transcriptional regulation (as shown in Figure 4C) which might prevent its translation in normal conditions (i.e. without overexpression, which could end up going over the miRNA-dependent threshold for translation). Regardless, we agree that non-functional would be more accurate and this has now been corrected in the new version of the manuscript (subsection “Cold temperature decreases TIM-L levels by two independent mechanisms”, first paragraph).

9) Figure 4C and subsection “Cold temperature decreases TIM-L levels by two independent mechanisms”, third paragraph. When the authors state that LUC reporter levels were high – compared to what? This suggests not much miRNA involvement for tim-sc. However as the endogenous tim-sc levels in the flies head are low at 25 °C (Figure 2C), does this mean that these low levels are little to do with miRNAs destabilising the transcript?

We thank the reviewers for raising this point. In Figure 4C, we compare the levels of luciferase obtained from the flies expressing the different reporters amongst themselves. We observed that the reporters including the *tim-M* or *tim-cold* 3’ UTR are significantly lower than the other reporters (those including *tim-sc* or *tim-L* 3’ UTRs). As all reporters are expressed at the same level (because the transgenes are inserted in the same genomic location) we assume that the differences are due to post-transcriptional control. Our hypothesis would be that the low levels of *timsc* at 25 °C are due to a lower retention of this intron at higher temperatures and not to an increased destabilization of the transcript by miRNAs or other posttranscriptional mechanisms. We have now rewritten this section to simplify and clarify this point (subsection “Cold temperature decreases TIM-L levels by two independent mechanisms”) as well as added a paragraph on the Discussion about this topic (seventh paragraph).

10) Subsection “Overexpression of the different TIM isoforms alters circadian behavior differently”, first paragraph. Is there also a corresponding advance in the M component in tim-sc – certainly looks like it – maybe also in tim-cold?

This is a good point. We had indeed quantified it and, although the trend is similar, we didn’t find statistically significant differences in the advance of the morning anticipation. We now added these results into Figure 5B and added a sentence in the first paragraph of the subsection “Overexpression of the different TIM isoforms alters circadian behavior differently”, to describe this.

11) Subsection “Elimination of tim-sc results in changes in tim processing and locomotor activity”, first paragraph. Some of the descriptions of the actograms are a bit subjective. I would have described the most obvious difference is the very strong siesta in the mutant at 29 °C. This is not mentioned. Also it appears that there is a reduction in M anticipation of lights on. One striking feature of Figure 6C is the lack of variation in the 40A profiles which makes it quite difficult to compare 40A with the control as the latter profiles seem so sloppy.

Reviewers are right. We have repeated this experiment multiple times but this lower variability in the 40A mutants compared to the controls has been consistent. We have modified the text to include the strong mid-day siesta phenotype exhibited by the mutants at 29 °C (subsection “Elimination of *tim-sc* results in changes in *tim* processing and locomotor activity, first paragraph).

12) The authors conclude that tim-M and tim-cold are strongly post transcriptionally regulated through miRNAs at all temperatures. While this matches with expression levels for tim-cold in Figure 2C, it is unclear with regard to tim-M which is relatively highly expressed at all temperatures often higher than the canonical isoform (18<25<29). This needs to be explained.

We thank the reviewers for pointing this out. Indeed, from the luciferase levels in flies overexpressing *luciferase* fused to the different *tim* 3’ UTRs we conclude that *tim-M* is also under strong post-transcriptional regulation. Additionally, in the AGO1-IP followed by small RNA sequencing we found that some of the miRNAs that have been predicted to bind to *tim-M* change in a temperature dependent manner (some being upregulated while other get downregulated with increasing temperature). This suggests to us that *tim-M* is under a strong post-transcriptional regulation at all temperatures. Although, as the reviewers mention, *tim-M* expression is very high at most temperatures, we argue that this strong posttranscriptional regulation does not conflict with the expression levels shown in Figure 2C as miRNAs can act by destabilizing mRNA and/or inhibiting translation. We have now acknowledged this in the Discussion (fourth paragraph).

[Editors' note: further revisions were requested prior to acceptance, as described below.]

The reviewers note that work linking generation of a 18C-specific, timeless splice isoform Tim-Sc to temperature-adaption of the circadian clock has been improved and tightened by several additional experiments. New analysis of Tim-Sc Per interactions and as well as phenotypic effects of tim-sc expression in per^0^ flies are consistent with that Tim-sc triggered, PER independent (but yet unspecified) mechanisms in temperature-adaptation of circadian clock.However, the manuscript requires extensive text revisions and some simple experimental additions before it can be considered acceptable for publication. The text is dense, difficult to read and appears to have been hastily assembled and needs a fair bit of work to make it more reader-friendly. It also contains several errors including an incorrect description of tim-sc which is fundamental to this manuscript and missing references. Further, it includes statements that are not supported by the data. Several very general claims about the tim splicing representing general cellular thermometers go way beyond the results – when a simple experiments exist to determine whether this may be true or not. Several, lines of changes are required. First, the manuscript needs to be carefully edited for clarity, grammar and accuracy to be easily accessible and comprehensive to the reader. And second, particularly given that the conclusions, while interesting are not decisive with respect to the mechanism by which Tim-Sc works, some key behavioral experiments (comments 12, 13) should also be conducted at colder temperatures (18 °C) to better test implications that the RNA modelling suggests for stability changes with temperature.

We thank the reviewers for their helpful comments and suggestions. We have now included behavioral and additional luciferase experiments at 18 °C. We have also carefully edited the manuscript to clarify several ambiguous aspects and added the missing references. We thank the reviewers for help us improving the manuscript.

Please find our point-by-point answer to the reviewers’ comments below.

Major comments:1) The opening sentence makes no sense – ' degradation of genes'? Maybe 'clock gene products'?

We thank the reviewers for bringing this up. This was our mistake. We have now modified the Abstract in the new version of the manuscript.

2) 'little is known…’ appears to be a bit harsh? Others who work in the field such as Ralf Stanewsky and Isaac Edery may disagree with this and perhaps their insights should be acknowledged.

We thank the reviewers for bringing this up. We have rephrased these statements in the new version of the manuscript.

3) Introduction, third paragraph. TIM is not the key factor, it's a key factor.

See below.

4) Introduction, third paragraph. tim is the key factor for light signalling' not 'TIM is the key pathway for conveying external information to the circadian system at the molecular level.' The authors are overplaying the role of TIM in all environmental inputs, light, temperature, social, etc. This is incorrect.

Answer to points 3 and 4: the reviewers are correct and we thank them for bringing this up. We have rephrased these statements in the new version of the manuscript.

5) Introduction, fourth paragraph. Majercak et al., 2004, should also be cited along with Collins et al., 2004.

We added this citation.

6) Introduction, fourth paragraph. The Edery group has done a lot of additional work on per splicing which has clarified somewhat how per splicing might impinge on thermal adaptation. Should be elaborated a little here.

The reviewers are correct. We have added a few new sentences to the Introduction.

7) Most figures – the authors need to state what the error bars represent -SEMS?

Thanks, the error bars do indeed represent the SEM. We have now corrected this in all the legends of figures and figure supplements.

8) Subsection “Temperature modulates tim alternative splicing”, first paragraph – there is a 3 fold increase at low temperature of per spliced, so tim is not the only clock gene with a significant thermal splicing pattern. Also Montelli et al., 2015, showed that at 10 °C per spliced reflects about 50% of total transcript and tim-cold about 90% of tim transcript at this temperature even though overall tim levels are low. It is surprising that this highly relevant study is not even mentioned or cited. We note that it is mentioned in the third paragraph of the Discussion in a different context but does not appear in the references.

We thank the reviewers for this point. We have described the patterns that we found more in detail now to address this issue. It is important to point out that cold induced isoforms represent more than 50% of all *tim* mRNAs at 18 °C and less than 10% at 25 °C, while the cold-induced isoform of *per* is no more than 30% of the transcripts at 18 °C. Montelli et al., 2015 had been cited a couple of sentences after when we described the *tim-cold* isoform and multiple times throughout the manuscript. However, we agree that adding a sentence clarifying that the patterns of temperature dependent *per* splicing could be amplified in a different temperature range could add to the manuscript. We have added this to the new version of the manuscript.

9) 'In contrast to the other isoforms which are generated by intron retention, tim-sc mRNA is generated by the usage of an alternative cleavage and polyadenylation site located within the intron 10 of tim which can only be used before this intron is spliced out.'

See below.

10) Apart from being ungrammatical, this sentence makes no biological sense. The authors suggest that the intron is spliced out. If so how can the STOP, cleavage and polyA sites be used? Surely this intron or part of it must also be retained?

Points 9 and 10: We thank the reviewers for noticing this mistake. Indeed, *tim-sc* is due to the retention of intron 10. Nevertheless, we wanted to emphasize that, unlike the order isoforms, *tim-sc* has an independent cleavage and polyA site. We have rephrased the sentence now to make it clearer.

11) 'As mentioned above, this isoform appears only at 18 °C, strongly suggesting that this isoform is responsible for the adaptation to cold temperatures'. This is an intuitive leap based on a correlation – no causation can be implied at this stage. If this was the case would not one expect higher levels of tim-sc and tim-cold in D. virilis at colder temperatures? So why is it 'surprising' that D. virilis does show higher levels of tim-cold at lower temperatures Doesn't the presence of tim-cold in all species at higher levels at colder temperatures argue that this is the conserved, important transcript for cold adaptation?

We thank the reviewers for this comment. We agree that correlation does not imply causation and have softened the statement. However, only the appearance of *timsc* correlates with differences in the day/night ration between 18 °C and 25 °C. We agree that the wider evolutionary conservation might suggest that *tim-cold* is important for general adaptation of the circadian system to temperature changes. However, we argue that *tim-sc* could be important for the flies to distinguish between 18 °C vs. 25 °C as from the strains analyzed in this study, only *D. melanogaster* and *D. simulans* (both of which express *tim-sc*) seem to have this defined temperature-dependent activity patterns between 18 °C and 25 °C. Although it is true that *D. virilis* also displays activity differences between these two temperatures, the pattern is the opposite (higher Day/night ratio of activity at 25 °C instead of lower). Additionally, *tim-cold* seems to be highly expressed in this strain, even at high temperatures (29 °C). In any case, we get the point and we modified the text to “important for” instead of “responsible for”, as indeed correlation does not imply causation at this stage.

12) Figure 5 – Why was the behavioral analysis nor also performed at 18 °C given that all this is done in a WT background? This needs to be done and included and the findings discussed.

We understand the point raised by the reviewers and have performed the experiment at 18 °C as suggested. However, we would like to point out that the main objective of that experiment was to determine the functionality of the proteins (in particular TIM-SC). In any case we included the new experiment in the new version of the manuscript.

13) Figure 6A to C. Again, might it not be useful to examine rescue of tim^01^ at lower temperature? Perhaps the different UTRs are more or less stable at different temperatures? This appears to be worth a shot perhaps given the last section on RNA structure?

We understand the point raised by the reviewers and have performed the experiment at 18 ºC as suggested. However, we would like to point out that all the *tim* overexpression constructs contain the same 3’ UTR (SV40, which is not controlled post-transcriptionally).

14) 'These results demonstrate that despite the smaller size, tim-sc encodes a fully functional TIM protein'. Why 'fully' with only 42% rescue?

We based our assessment on the fact that the canonical TIM-L protein rescues at a similar extent. We interpret this partial rescue to the use of the GAL4 system and not the endogenous promoter to drive TIM expression. In any case, we have softened this statement and added a few sentences to the Discussion.

15) 'Control flies presented two different responses when transferred to 18 °C: a delay of the evening component as well as a decreased night activity in comparison to the flies maintained at 25 °C '(Figure 6C, 6D and 6E).' Do the authors mean Figure 7C, D, E.

We thank the reviewers for bringing this up. We have corrected the mistake in the new version of the manuscript.

16) 'In addition, the activity patterns of 40A flies are strikingly similar at the different temperatures.' Not really, and no more similar than controls. If one examines panels 7D and 7E the two phenotypes change more in 40A than in controls.

We thank the reviewer for bringing this up. We have corrected accordingly. As we explain in the text, we interpret the still high day/night activity at 18 °C to the rerouting of transcription towards the *tim-cold* splicing variant (Figure 7F).

17) 'Last but not least, our data suggest that tim alternative splicing might act as a thermometer for the cell.' This is a very general statement, not even limited to the circadian clock. If this is the case then other temperature dependent phenotypes should also be affected by the tim-splicing mutants. Why don't the authors check one of these out, for example the temperature at which temperature-sensitive mutants become paralysed e.g. shits, parats etc? Simple enough to do but unlikely to support the authors general claim.

We understand the point raised by the reviewers. We have modified the statement to make it specific to the circadian clock.